# Resolving Lexical Bias in Edit Scoping with Projector Editor Networks

## Abstract

Weight-preserving large language model editing techniques rely heavily on scoping mechanisms that determine when to apply edits to the base model. These mechanisms typically use distance functions in the representation space. However, we demonstrate that distance-based scoping functions struggle with strong lexical biases, leading to issues such as applying edits to irrelevant prompts with overlapping words. This paper presents Projector Editor Networks for Model Editing (PENME), a principled approach that learns the optimal representation space for scoping using contrastive learning. Specifically, PENME forms a disentangled representation space that facilitates precise localization of edits by maintaining substantial distance between irrelevant prompts while preserving proximity among paraphrases. In our empirical study, we show PENME achieves state-of-the-art model editing results while being more computationally efficient during inference and adaptable across different architectures.

## 1 Introduction

Large Language Models (LLMs) have demonstrated tremendous success in solving a diverse range of natural language processing tasks (Devlin et al., 2018; Liu et al., 2019; Touvron et al., 2023b; Radford et al., 2019). Despite their successes, LLMs are fallible. One reason for this discrepancy is the noisy and imperfect nature of the data used for training (Zhu et al., 2020). As the world evolves, new information requires updates to the models e.g. the prime minister of a country may change over time. Models trained on outdated data are prone to making factual errors.

Periodically retraining LLMs is one potential solution, but risks degrading performance, often requiring training from scratch to maintain previous capabilities (Luo et al., 2023; Wang et al., 2023b). Retraining requires significant computational resources, investment of time, data, and skilled labor.

To alleviate this, model editing was proposed to perform sample and compute efficient knowledge updates. There are two primary model editing design paradigms: ***weight-modifying*** and ***weight-preserving***. ***Weight-modifying*** approaches directly update the model's parameters to integrate new information (Meng et al., 2022a;b). Although these approaches are sample efficient, they demand substantial compute resources for training (Yu et al., 2024) and result in catastrophic forgetting (Gupta et al., 2024a) the full impact of which is difficult to fully determine (Rosati et al., 2024).

In contrast, ***weight preserving*** approaches maintain the original model parameters while employing additional components to reflect updated knowledge, thereby avoiding catastrophic forgetting (Hartvigsen et al., 2024; Yu et al., 2024). These methods either utilize user inputs and memory storage before a forward pass of the model (***pre-input***) or during a forward pass (***post-input***). **Pre-input** approaches (Mitchell et al., 2022; Zheng et al., 2023) rely on retrieving relevant contexts for model editing, this process requires additional components such as retrievers and counterfactual models which make them computationally intensive. **Post-input** mechanisms (Hartvigsen et al., 2024; Yu et al., 2024) involve adapter-based editing techniques that incorporate components within the model's computational pathway, modifying its output to reflect edited information.

A core component of adapter-based techniques is a vector similarity scoping mechanism that utilizes model representations and memory codebooks containing vector representations to determine when to utilize computational paths associated with a given edit. However, we observe that adapter-based techniques struggle with handling paraphrases and often misfire on irrelevant prompts with

Figure 1: An illustration of lexical dominance in embeddings: a) shows setting the similarity threshold low (illustrated with the circle) which results in failing to edit paraphrases. b) shows setting the similarity threshold high resulting in misfires with irrelevant prompts. c) illustrates the representation space that our projector network learns.

similar lexical content. This is in line with the findings of Dumpala et al. (2024) who examine the impact of lexical diversity on model representations for semantically equivalent texts and showed that their representations often exhibit divergence despite semantic equivalence. For example, the representation for "The twin city of Pittsburgh is" may exhibit greater similarity to "The twin city of Portsmouth is" than to its paraphrase "Pittsburgh is a twin city of".

Our work uniquely characterizes this problem (§6.1) leading to our novel finding that for model editing, lexical factors predominantly shape model representations. Figure 7 shows that 58% of edits from the Counterfact dataset (Meng et al., 2022a) were closer to unrelated neighbours than the edit paraphrases using representational similarity measures. This result is misfiring of the scoping mechanism i.e. inappropriately applying an edit. This phenomenon creates a trade-off between effectively executing the correct editing mechanism on paraphrases and preventing misfires on irrelevant prompts. A low distance threshold, which controls the scoping mechanism, reduces misfires but impedes paraphrase execution, while a higher threshold enhances paraphrase performance but increases misfire risk as illustrated in Figure 1.

In this work, we introduce Projector Editor Networks for Model Editing (PENME), an advancement over previous adapter-based weight preserving model editing by explicitly targeting the lexical bias problem in these scoping mechanisms. PENME comprises of two key components: a projector network and a similarity-based retrieval system. The projector network is a compact, two-layer neural network trained independently using contrastive learning to disentangle projection space such that paraphrases of edits demonstrate proximity, while irrelevant prompts, both with and without similar lexical overlaps, are farther away. Based on the outputs of the projector network, a memory-based retrieval system facilitates efficient edit retrieval. This approach effectively addresses the aforementioned challenges, while maintaining computational efficiency and ensuring compatibility with both encoder- and decoder-based architectures.

Our contributions are as follows: **(1)** We demonstrate that representations extracted across layers from various LLMs exhibit lexical dominance, showing a bias towards token overlap which introduces significant challenges for adapter-based model editing techniques. **(2)** We propose a projection network that maps the model's representation space to a new representation space where lexical dominance is minimized. **(3)** We integrate our projection network in an adapter and memory-based retrieval scheme for model editing, demonstrating high efficacy for paraphrase execution (generalization), preventing misfires on irrelevant prompts (locality) and it's generalization to unseen edits, paraphrases, and neighbours. The proposed projection network is a novel solution to the problem in hand. Moreover, It has broader impact to other application areas relying on representation similarities such as retrieval augmented generation, however, it is out of the scope of this paper.

## 2 RELATED WORK

***Weight-modifiying*** approaches typically rely on the localization hypothesis (Miller et al., 2016; Geva et al., 2020) in the transformer architecture which conjectures pointwise feed-forward components function similar to a key-value memory for information retention within a LLM (a hypothesis which has recently been criticised w.r.t model editing in Hase et al., 2023). Meng et al. (2022a) identifies

salient neurons within the feed-forward layers, facilitating targeted updates to effect the desired edits using causal analysis. Similarly, Li et al. (2024) investigates the role of multi-headed attention, in conjunction with feed-forward layers, for model editing. Mitchell et al. (2021) uses a hypernetwork to predict new weights for the model by using a low-rank decomposition of the weight matrices of different layers. The goal is to edit information in the model parameters without impacting unrelated information.

**Weight Preserving: pre-input** approaches depend on extracting and processing relevant edit information before the input is processed by the main model. For example, SERAC (Mitchell et al., 2022) employs a memory-based model editing strategy augmenting a primary LLM with two additional models and memory storage. The supplementary models determine scope-of-edit and perform counterfactual reasoning. Retrieval-augmented (RAG) techniques like IKE (Zheng et al., 2023) leverage similarity-based retrieval to extract and rank edit demonstrations from memory and use in-context reasoning to perform edits.

**Weight preserving: post-input** rely on the model's internal representations to implement scoping mechanisms (mechanisms which determine whether a specific edit applies for the current input) and employ a playback mechanism that triggers the model to generate modified outputs. For example, GRACE (Hartvigsen et al., 2024; Yu et al., 2024) operate an in-model adapter approach. These approaches employ a codebook or memory storage system to maintain model representations of edits as clusters. They utilize a vector similarity-based retrieval mechanism to generalize edit paraphrases and constrain irrelevant or neighbouring prompts. Initial cluster sizes are deliberately restricted to mitigate interference from neighbouring prompts and ensure that only paraphrases of the edit prompt are mapped within the cluster. However, this design necessitates continuous cluster resizing, as new edits with similar semantic and lexical properties may fall within an existing edit cluster. Furthermore, the initially small cluster radius necessitates the storage of multiple edit paraphrases in the memory codebook to achieve effective generalization, potentially leading to increased memory consumption. The major difference in the approaches is that Hartvigsen et al. (2024) uses memory playback vectors while Yu et al. (2024) uses LoRA blocks (Liu et al., 2024) for the generation process. An alternative editing method involves enhancing the feedforward layer (FFN) within a transformer block by incorporating additional neurons to facilitate the desired modifications. Huang et al. (2023) introduce a single neuron per output token for edited information of single edit. In this framework, each neuron, or a group of neurons, is specifically trained to activate solely for a particular edit, thus adjusting the model's output to produce the altered information.

Cluster-based similarity systems like GRACE and MELO Hartvigsen et al. (2024) and Yu et al. (2024) rely on concept separability within the representation space to manually maintain keys in their codebooks. However, our analysis reveals that lexically similar prompts cluster closer to edits than their paraphrases, heightening the risk of system failure as can be seen from figure 1 and 3. Moreover, their cluster based design necessitates storing edit paraphrases as codebook entries for effective generalization which increases retrieval latency. PENME overcomes this limitation by learning a projection space that enhances representation structure, enabling more effective organization of keys for faster and more accurate retrieval. Furthermore, PENME consistently outperforms both weight-preserving and weight-modifying methods across various architectures, underscoring its adaptability and efficacy.

## 3 PROBLEM SETTING: MODEL EDITING

The objective of model editing is to alleviate the need for complete retraining by updating the model under the following conditions (1) sample efficiency: update the model with the fewest number of samples possible, (2) compute efficiency: train a small portion of the model only, (3) minimal impact: make as small of an impact on unrelated behaviour as possible (for adapter-based approaches this means preventing misfires on irrelevant prompts) and (4) ensure generalization: maintain accurate paraphrase behaviour (for adapter-based approaches this means retrieval of the correct edits).

The aim is to modify the behaviour of a model $M$ on a dataset $D = [d_1, d_2, d_3...d_n]$ where the sample $d_i$ is the tuple $(x_i, y_i, [p_{i1}, p_{i2}...], [nb_{i2}, nb_{i2}, ..])$. In this tuple, $x_i$ is the edit prompt, $y_i$ is the new output tokens, $p_{1..n}$ are the edit paraphrase prompts, $nb_{1..n}$ are *neighbours* or *neighbourhood prompts* these are lexically and semantically similar prompts but ones where the underlying model generation should not change. To be successful in the model editing task, the edited model, $M_{edited}$,

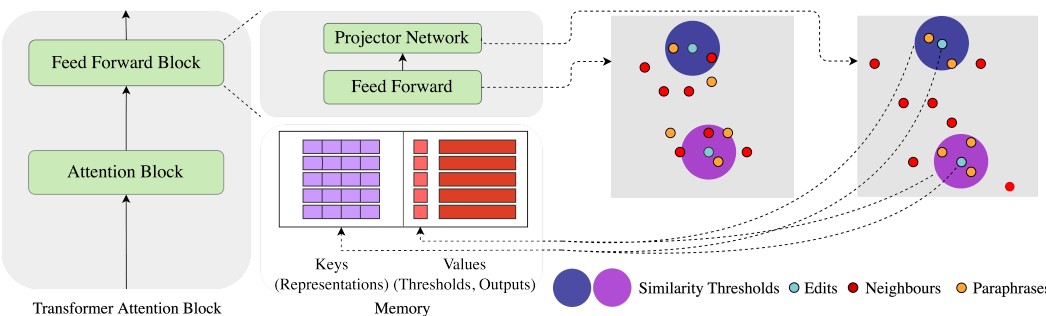

Figure 2: **PENME** uses a projection network that interfaces with the pointwise feed-forward layer output in a selected transformer block. This projection network, coupled with key-value codebook storage, acts as a scoping mechanism by comparing projection outputs with codebook entries. This mechanism determines whether the current input relates to a specific edit or should pass through the model unmodified.

should generate new target tokens $y_i$ for a specific input $x_i$ (Edit Success) and its related paraphrases $p_{1..n}$ (Generalization), while maintaining the model's behaviour on semantically unrelated prompts $nb_{1..n}$ (Locality). The following metrics illustrate how these factors are typically operationalized (see for example Yao et al., 2023; Yu et al., 2024; Hartvigsen et al., 2024; Gupta et al., 2024b).

**Edit Success (ES):** The proportions of edits that the model is able to recall or generate correctly. This metric has also been called efficacy, reliability, and edit score and is denoted as:

$$M_{edited}(x_i) = y_i, \, \forall (x_i, y_i) \in d_{1:n} \tag{1}$$

**Locality:** The proportion of prompts concerning neighbouring entities unrelated to the edit for which the model generates the same outputs prior to editing, this is also described as specificity, neighbourhood success, retain rate and neighbourhood score. Denoted as:

$$M_{edited}(nb_{ij}) = M(nb_{ij}), \, \forall nb_{ij} \in nb_i, \forall nb_i \in d_{1:n}. \tag{2}$$

**Generalization:** The proportion of paraphrases for which the model is able to recall or generate the correct edited information, also described as paraphrase success, paraphrase score:

$$M_{edited}(p_{ij}) = y_i, \, \forall p_j \in p_i, \forall (p_i, y_i) \in d_{1:n}. \tag{3}$$

**Score:** The general score is the mean of the above three metrics used for benchmarking.

## 4 PROJECTOR EDITOR NETWORKS FOR MODEL EDITING (PENME)

PENME is a retrieval-based editor that leverages an adapter module which is integrated alongside pointwise feed-forward layers within an attention block of a pre-trained Large Language Model (LLM). By introducing this additional component rather than altering the original model weights, PENME enables the integration of new information while preserving the LLM's initial capabilities.

PENME illustrated in Figure 2 consists of two components, (1) **Projection Network** ($M_{proj}$): this component projects model activations at a specific layer $M_l(x)$ into a distinct representation space. (2) **Key-Value Codebook** that stores model activations at layer $M_{proj}(M_l(x))$ as keys and corresponding values containing a learned similarity threshold ($\delta$) and the new associated output information $y_i$. It should be noted that instead of output information, vectors can also be stored as values which facilitate playback approaches such as vector playback (Hartvigsen et al., 2024) and LoRA blocks based playback (Yu et al., 2024). Output retrieval and playback are compatible with all transformer-based model architectures.

## 4.1 PROJECTION NETWORK

The projection network $M_{proj}$ is a small feed-forward neural network trained via contrastive learning (Hadsell et al., 2006) with additional constraints, defined by the following loss function:

$$\mathcal{L}(\vec{x_i}, \vec{z}) = (1 - t)\frac{1}{2}||\vec{x_i} - \vec{z}||_2^2 + t\frac{1}{2}[\max(0, m - ||\vec{x_i} - \vec{z}||_2)^2]$$
$$t = \begin{cases} 1 \text{ if } \vec{z} \leftarrow \vec{p_{ij}}, \\ 0 \text{ if } \vec{z} \leftarrow \vec{n_{ij}} \vee \vec{x_l} \end{cases} \tag{4}$$

where $t$ is the target $\{0, 1\}$ which is 0 when the training pair is $\{x_i, p_{ij}$ (edit,paraphrase) and 1 when $\{x_i, nb_{ij}\}$ (edit,neighbour) or the inter-edit (or edit-to-edit) pair $\{x_i, x_l\}$ , $m$ is the margin which pushes $\vec{n_{ij}}$ at least $m$ distance away from $\vec{x_i}$. The projection network is trained such that for all samples in a dataset, edits $x_i$ and edit paraphrases $p_{ij}$ are close together while edits $x_i$ and neighbours $nb_{ij}$ are distanced in the projection space i.e. $||\vec{M_{proj}}(\vec{M_l}(x_i)) - \vec{M_{proj}}(\vec{M_l}(p_{ij}))||_2 \ll ||\vec{M_{proj}}(\vec{M_l}(x_i)) - \vec{M_{proj}}(\vec{M_l}(nb_{ij}))||_2$. Training is performed by sampling pairs at random and $\vec{z}$ in the loss function above is assigned based on the pair category we discussed earlier. The conventional contrastive training for the projection network results in a suboptimal solution. The inherent lexical and semantic similarities among edits increase the probability of certain edit paraphrases exhibiting greater proximity to other edits. This phenomenon can lead to erroneous paraphrase-edit associations during execution, potentially triggering inappropriate edit operations. To mitigate this issue, we propose an enhanced approach that incorporates an additional constraint to maximize $||\vec{M_{proj}}(\vec{M_l}(x_i)) - \vec{M_{proj}}(\vec{M_l}(x_l)||_2$ where $x_l$ is sampled from other edits in the dataset. This results in increasing the inter-edit distances $||\vec{M_{proj}}(\vec{M_l}(x_i)) - \vec{M_{proj}}(\vec{M_l}(p_{ij}))||_2 \ll ||\vec{M_{proj}}(\vec{M_l}(x_i)) - \vec{M_{proj}}(\vec{M_l}(x_l))||_2$. This novel modification serves to expand the overall projection space, thereby reducing the likelihood of misclassification. The number of edit-to-edit pairings is determined by the similarity between edits, which is controlled by a hyperparameter $\Phi$.

The compact architecture of the projection network enables it to be trained on GPUs with limited memory capacity, irrespective of the underlying model's scale. We provide the details of implementation, data construction and training in Appendix A.

## 4.2 KEY-VALUE MEMORY

The key-value memory is designed to store edits and their corresponding outputs. For each edit, representations are generated by passing the input $x_i$ through the model and the projection network. This is denoted $act_{x_i} = M_{proj}(M_l(x_i))$. These representations are then stored as keys $k_i \in K$ in the memory and are utilized during runtime in a similarity-based retrieval system to access the relevant edit. The memory value $v_i \in V$ consists of the edited information along with a similarity threshold. The threshold serves as a scoping mechanism. For a given input prompt denoted $pt$, euclidean distance $|| \cdot ||_2$ is computed with all keys in the memory. From the computed distances, we determine if the input prompt $pt$ is relevant to the edited memory value $v_i^\mu$ and its corresponding threshold $v_i^\delta$. This is expressed as:

$$\underset{k_i, v_i}{argmin} \; ||act_{pt} - k_i||_2$$
$$s.t. ||act_{pt} - k_i||_2 < v_\delta^i \tag{5}$$

If the prompt $pt$ is deemed relevant based on the equation 5, the output information of the edit is retrieved from memory $v_i^\mu$. Otherwise, the typical model output $M(pt)$ is employed.

Initial experimental findings regarding the threshold reveal that unseen test paraphrases typically demonstrate greater distance than the average seen training paraphrases, while the inter-paraphrase distances within the training set exhibit variation across edits. In contrast, unseen test neighbours generally show closer proximity to edits compared to the nearest seen training neighbour, this effect is illustrated in greater detail in Appendix B. To determine an appropriate threshold that defines the scope of an edit, we investigate various data-driven thresholding schemes based on the training data.

1. $Max(||\vec{x} - \vec{p_{ij}}||_2) + \tau$, setting $\tau$ distance away from max paraphrase distance

2. $Min(||\vec{x} - \vec{nb}_{ij}||_2) - \tau$, setting $\tau$ distance below min neighbour value

The selection between the two alternatives is contingent upon the specific aspect of adjustment that is prioritized. Option 2 maintains locality by preserving all training neighbours, while Option 1 assures that all training paraphrases will be operational regardless of the $\tau$ value selected. In Option 2, the final threshold, after adjustment with $\tau$ for certain edits, is set closer to the farthest paraphrase. We opt for Option 1 in our experiments, as it guarantees a full edit success rate.

**Edit Removal and Scalability**: The scoping mechanism employed by Hartvigsen et al. (2024); Yu et al. (2024) requires multiple paraphrases added to the codebook to improve generalization. To enhance efficiency, merging operations are performed on nearby edits that produce identical outputs. However, the efficacy of this consolidation is dataset-dependent; for example, zsRE demonstrates a high frequency of similar edit outputs, enabling a significant reduction in codebook entries. For example, 1000 edits on zsRE requires 658 entries in total but for Counterfact 1682 entries are needed just for 300 edits. The combination of this consolidation process and the potential for edits to be closely related in vector space leads to overlapping cluster radii, necessitating cluster size reduction. This inadvertently results in the removal of certain edits. Thus edits can be forgotten. In contrast, our method exhibits linear scaling with respect to the number of edits in the worst-case scenario where each edit produces a unique output as exhibited in Appendix C. This characteristic allows for more rapid edit retrieval compared to the aforementioned approach. Furthermore, our method facilitates straightforward edit removal or updates, offering enhanced flexibility in edit management.

## 5 EXPERIMENTAL SETUP

We assess the performance of PENME across a spectrum of transformer-based LLMs including Text-to-Text Transfer Transformer (specifically T5-small) (Raffel et al., 2020), Llama-2-7b (Touvron et al., 2023a) and GPT2-XL (Radford et al., 2019). We compare PENME with We compare PENME with GRACE and MELO, as these are weight-preserving approaches that closely align with our methodology. Additionally, we include MEMIT and SERAC in the evaluation, as they high performing techniques in model editing alongn with a baseline that uses PENME's thresholding system. The baseline is refered to as Defer. . Working details of the methods and hyperparameters are provided in Appendix D.1. To select the optimal layer to introduce the PENME adapter, we utilize the methodology outlined in section §6.1 and incorporate PENME in the second layer for all LLMs. To determine the optimal threshold for each edit, we systematically vary the $\tau$ parameter in Equation equation 2 across a range of 0.05 to 0.20.

**Dataset**   The zsRE dataset (Levy et al., 2017) and the Counterfact dataset (Meng et al., 2022a) are the commonly used model editing datasets for evaluation. The zsRE dataset consists of an edit prompt along with several paraphrased versions of that prompt. To evaluate the impact of edits on unrelated knowledge, neighbourhood prompts are sourced from the NQ dataset (Kwiatkowski et al., 2019), which offers a wide range of user query questions. In contrast, Counterfact has similar edit and paraphrase prompts but employs a more nuanced approach to neighbouring prompts. It includes prompts that are similar to the edit prompt in both semantic nature and lexical structure. This differs significantly from zsRE, where the neighbouring prompts are neither semantically nor lexically related to the edit prompt. Moreover, zsRE has a lower spectrum of subjects, relationships, and linguistic variations. This structural difference between the datasets has important implications for evaluation. In zsRE, the lack of semantic or lexical relationships between the edit prompt and its neighbours allows weight-preserving approaches to achieve high locality scores with relative ease. The enhanced complexity of Counterfact renders it a more robust benchmark for evaluating editing mechanisms. Dataset processing and training data construction details for both datasets are provided in Appendix D.2.

## 6 EVALUATION

In this section, we present evidence of lexical dominance, the results of PENME in achieving separability of unrelated neighbours and paraphrases, and comparison with other methods.

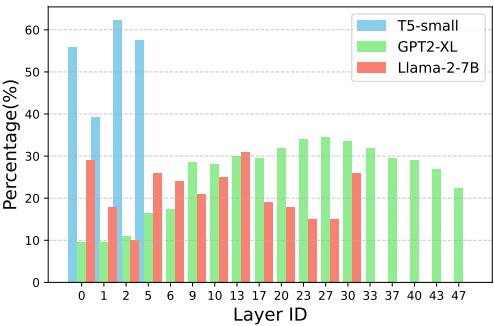 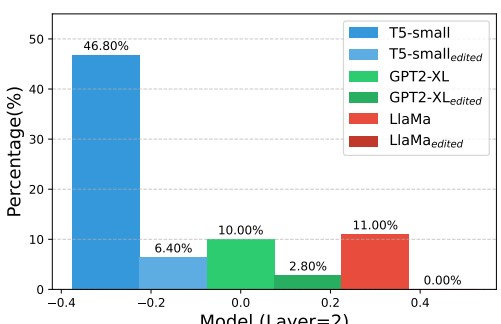

Figure 3: Percentage of samples *where edits are closer* to unrelated neighbours as compared to paraphrases in the representations space of different models across various select layers. T5-small, GPT2-XL and Llama-2-7b have 6, 32, 48 layers respectively. The full figure for all layers can be found in Appendix E.1

Figure 4: Percentage of samples *where edits are closer* to unrelated neighbours as compared to paraphrases in the representation space of different models and projector networks. Lower percentages indicate better performance.

## 6.1 LEXICAL DOMINANCE

To examine the lexical dominance of representations, we randomly sampled 500 entries from the Counterfact dataset (see §5). For each entry, we created triplets consisting of an edit prompt, a randomly sampled paraphrase prompt and a neighbouring prompt with high lexical overlap $(x_i, p_i, n_i)$. These triplets are fed into various models, and representation vectors $(\vec{x_i}, \vec{p_i}, \vec{n_i})$ from the feed-forward block of each layer $l$ are extracted for all samples. We select either averaged token representations or dedicated sentence representation, based on whether a given model offers a specific token for sentence-level representation. Following extraction, we calculate two sets of pairwise Euclidean distances: (1) Between edit representations and paraphrase representations: $||\vec{x_i} - \vec{p_i}||_2$ (2) Between edit representations and neighbour representations: $||\vec{x_i} - \vec{nb_i}||_2$. We then compare these distances to determine if neighbours are closer to the edits than the paraphrases $||\vec{x_i} - \vec{p_i}||_2 > ||\vec{x_i} - \vec{nb_i}||_2$. Figure 3 displays the percentage of samples where neighbours *were closer* to the edits.

The findings reveal an intriguing pattern: except for the first layer in most models, the early layers demonstrate a reduced percentage of samples where neighbours are closer to edits than paraphrases. However, the trend shifts as we progress through the model's depth. In the mid-layers, this percentage begins to ascend once more, only to descend slightly towards the final layers, albeit with subtle fluctuations among them. We hypothesize that in the initial layers, token-specific information remains largely isolated. However, as the input traverses deeper into the model, guided by repeated attention mechanisms, this information becomes amalgamated across tokens. Moreover, repeated normalization as demonstrated by Takase et al. (2022) results in smaller changes in weights of an LLM leading to embedding vectors in the final layers being similar thus only subtle fluctuations are seen in the percentages.

These results indicate why there is a significant chance of misfire in **post-input** methods. This also provides a systematic approach for identifying the optimal layer to introduce PENME integration, by elucidating the regions within the model's architecture where lexical dominance exhibits minimal influence. Although the projector network approach can be generalized across all layers, as demonstrated in Appendix E.2, it is advantageous in terms of training time to integrate at points of minimal influence.

## 6.2 DISENTANGLED PROJECTION SPACE

In this section, we validate our proposed projection network in its ability to learn a generalized disentangled representation space where paraphrases are closer to edits as compared to neighbours. We sample 1500 tuples of edits $(e_i)$, paraphrases $p_i$, and their unrelated neighbours $nb_i$ $(e_i, p_i, nb_i)$ from the Counterfact dataset with accompanying input prompts $x_i$ and split them into train and test

| Method | Model | CounterFact | | | | zsRE | | | |
|---|---|---|---|---|---|---|---|---|---|
| | | ES | Loc | Para | Score | ES | Loc | Para | Score |
| PENME | T5-small | **1.000** | 0.787 | **0.808** | **0.865** | **1.000** | **0.941** | 0.913 | **0.951** |
| | Llama-2-7b | **1.000** | 0.869 | **0.906** | **0.925** | **1.000** | **0.987** | **0.966** | **0.984** |
| | GPT2-XL | **1.000** | 0.847 | **0.875** | **0.907** | **1.000** | **0.957** | 0.940 | **0.966** |
| MELO | T5-small | 0.850 | 0.800 | 0.037 | 0.562 | 0.990 | 0.640 | **0.986** | 0.872 |
| | GPT2-XL | **1.000** | **1.000** | 0.020 | 0.673 | **1.000** | 0.004 | **1.000** | 0.668 |
| GRACE | T5-small | **1.000** | **0.860** | 0.140 | 0.667 | **1.000** | 0.730 | **0.993** | 0.907 |
| | Llama-2-7b | **1.000** | **0.997** | 0.002 | 0.666 | 0.120* | 0.00* | 0.579* | 0.233* |
| | GPT2-XL | **1.000** | 0.996 | 0.003 | 0.666 | 0.993* | 0.019* | 0.017* | 0.343* |
| SERAC | T5-small | 0.017 | 0.526 | 0.010 | 0.184 | 0.017 | 0.526 | 0.010 | 0.184 |
| | Llama-2-7b | 0.992 | 0.372 | 0.651 | 0.672 | **1.000** | 0.114 | 0.357 | 0.490 |
| | GPT2-XL | 0.947 | 0.669 | 0.408 | 0.675 | 0.474 | 0.003 | 0.811 | 0.429 |
| MEMIT | Llama-2-7b | 0.147 | 0.149 | **1.000** | 0.432 | 0.402 | 0.002 | **1.000** | 0.468 |
| | GPT2-XL | 0.785 | 0.788 | 0.502 | 0.692 | 0.214 | 0.000 | **1.000** | 0.405 |
| FT | T5-small | 0.955 | 0.000 | 0.450 | 0.468 | 0.017 | 0.526 | 0.010 | 0.184 |
| | Llama-2-7b | 0.404 | 0.393 | 0.417 | 0.405 | 0.569 | 0.020 | 0.746 | 0.445 |
| | GPT2-XL | 0.968 | 0.851 | 0.395 | 0.738 | 0.608 | 0.005 | 0.889 | 0.501 |

Table 1: A comparative analysis of PENME and recent model editing methods on 2000 edits from the Counterfactual dataset and 1000 edits on zsRE. The metrics are Edit Sucess (ES), Locality (Loc) and Paraphrase Generalization (Para). *indicates only a subset of 100 is computed.

sets of 1000 and 500 samples respectively. We use the training set to train the projector network using model representations from layer 2 of each model. To evaluate the network's performance, we compare two types of test representations: the original model representations $M_l(x_i)$ where $x_i$ is the input prompt and the projected representations $M_{proj}(M_l(x_i))$. This comparison uses the experimental method described earlier, allowing us to determine whether the projection network successfully learns to create a representation space with the desired properties.

The results presented in Figure 4 demonstrate that the projector network, despite not being exposed to these specific samples during training, effectively learns to distance lexically similar but unrelated neighbours in comparison to paraphrases. A two-dimensional PCA visualization of the representation space, illustrating this phenomenon, is provided in Appendix F.2.

For data pairs where neighbours are closer to edits than paraphrases, T5-small exhibits a significant decrease from $46\%$ percent to $6.4\%$. Similarly, GPT2-XL shows a reduction of over $7\%$, and Llama-2-7b drops to $0\%$, indicating perfect separability of neighbours and paraphrases.

## 6.3 MODEL EDITING RESULTS

Table 1 presents the comparative results of PENME and recent model editing methods for 2000 edits on the Counterfact dataset and 1000[1] edits on zsRE. PENME demonstrates a highly stable performance across editing metrics as compared to other model editing approaches. In particular, PENME shows high efficacy on locality and generalization compared to other model editing approaches and has more stable performance across the different models.

GRACE, similar to PENME, demonstrates high edit success rates due to its inherent design. However, its generalization scores compared to PENME were markedly low, suggesting poor performance on edit paraphrases post-editing. GRACE achieved the highest locality scores, with T5-small at 0.92 and Llama-2-7b nearly perfect at 0.997. The substantial difference between locality and generalization scores can be attributed to GRACE's use of a very low distance threshold, resulting in poor performance on paraphrases but successfully avoiding neighbouring prompt spillover into edits.

---

[1] *Due to system implementation issues with GRACE on EasyEdit (Wang et al., 2023a), we were only able to compute a 100 sample subset for results with a *.

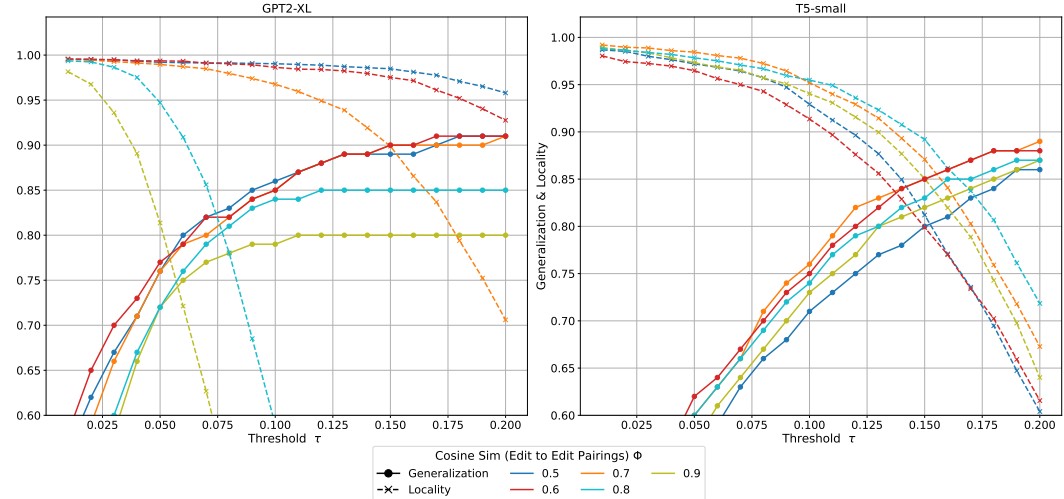

Figure 5: Shows the trade-off between generalization and locality performance across different hyperparameter settings. The distance threshold $\tau$ varies from 0.01 to 0.2 (0.01 increments and $\tau$ is normalized by 100), while the edit-pairing similarity threshold $\phi$ ranges from 0.5 to 0.9 (0.1 increments). Higher $\phi$ values enforce stricter edit similarity requirements. The results showcase the effect of hyperparameter tuning on the projector network's learning capacity and overall performance.

SERAC also achieves a high edit success but shows a low and mixed performance for generalization and locality across models. For T5-small, the approach does not work well as SERAC uses logically entailed facts "prompt: TRUE or FALSE" to determine the scope, the original work uses a T5-large which is significantly better at reasoning.

For GPT2-XL, MEMIT demonstrates moderate effectiveness, achieving an edit success rate of 0.785 and a locality score of 0.788. In contrast, when applied to Llama-2-7b, both the edit success and paraphrase success rates are relatively low, although the locality score remains high. This discrepancy is likely attributed to challenges arising from MEMIT's training on the Llama-2-7b model.

# 7 ABLATIONS

## 7.1 GENERALIZATION AND LOCALITY

To demonstrate the trade-off between generalization and locality, we conducted an ablation study by varying the alpha parameter, which modulates the similarity threshold defining an edit's scope. Figure 5 presents the results for GPT2-XL and T5-small. The trends observed for GPT2-XL and Llama-2-7b are similar. Therefore, for clearer visualization, we present the detailed results for Llama-2-7b separately in Appendix F.1. Setting a low $\tau$ value achieves near-perfect locality but poor generalization. As we incrementally increase the threshold, generalization improves while locality declines gradually. Generalization values either plateau for larger models (Llama-2-7b, GPT2-XL) or continue to increase for smaller models (T5-small). Each model exhibits an optimal threshold where generalization and locality are balanced; these thresholds can be adjusted to suit specific use cases e.g. high locality to ensure no degradation in the original model.

Figure 5 also illustrates the impact of varying the similarity threshold for edit-to-edit pairings in the training dataset on the projector network's learning. Edit-to-edit pairings $\phi$ which move edits farther away from each other are central to training a robust projector network. For T5-small, training remains largely stable across all thresholds, with optimal performance at the midpoint (0.70); deviations from this threshold result in decreased overall performance balance between generalization and locality. In larger models, threshold selection proves critical, as inappropriate values can lead to training instability, causing early plateauing of generalization and rapid decline in locality. The threshold value for edit-to-edit pairings $\phi$ significantly impacts training stability and performance. Higher thresholds, such as 0.75, result in fewer pairings and lead to unstable training for both Llama-

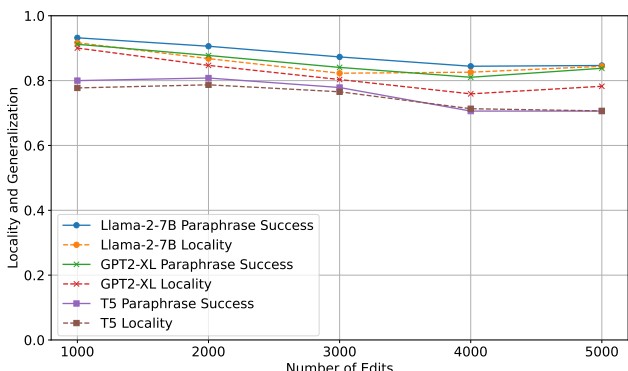

Figure 6: PENME's performance in terms of Locality (dotted line) and Generalization (continuous line) across varying numbers of total edits performed.

2-7b and GPT2-XL models, ultimately resulting in poor performance. Conversely, lower thresholds, exemplified by 0.6, increase the number of pairings and enhance training stability.

## 7.2 Scaling Edits

We evaluate the projection network's stability under varying numbers of edits using incrementally larger training sets ranging from 1000 to 5000 edits, with 1000-edit increments per training session. The results of the experiment are shown in Figure 6. Projector network trained on representations from T5-small demonstrates lower overall performance in generalization and locality compared to other models. We hypothesize that this under-performance may be attributed to either the model's smaller size, resulting in less robust learned representations, or the fact that it was trained on a more limited dataset relative to larger, more recent models. Projection networks trained on Llama-2-7b and GPT2-XL representations exhibit comparable performance levels. Both models show a slight decrease in generalization and locality performance as the number of edits increases from 1000 to 2000, with minimal decline after that. For T5-small the performance is relatively stable up to 3000 edits after which a more pronounced decline is observed.

Examination of projection network behaviour reveals interesting patterns in generalization and locality failures based on the varying distances between training edits and their respective paraphrases and neighbours after the training of the projector network. The varying distances result in different thresholds for each edit, which can cause errors when the closest edit to a neighbouring example has a high threshold. To quantify these observations, we employed ROUGE scores in a comparative study of generalization outcomes. Appendix G provides this analysis, offering insights into the nuances of the learned projection space.

## 8 Conclusion

In this paper, we proposed PENME an adapter-based model editing approach that utilizes a projection network trained via contrastive learning. PENME explicitly targets the lexical bias present in representations that causes misfiring of editing scope. Moreover, it used a memory-based storage system alongside the scoping mechanism for efficient edit retrieval. Empirical evaluations demonstrated PENME's superior performance across varying levels of task complexity. On the zsRE dataset, It achieved impressive generalization and locality scores exceeding 0.90. Notably, when assessed on the more challenging Counterfact benchmark, the system maintained robust performance, attaining scores above 0.80 for both generalization and locality metrics. This performance on Counterfact is particularly significant given the benchmark's increased difficulty, underscoring PENME's efficacy. In the future, we plan to assess whether a projector, pretrained on a large dataset to maximize semantic information, could be used as a plug-and-play solution without requiring additional training. Moreover, we intend to expand PENME to encompass more scenarios, including long-form generation.

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

## A    DATA CONSTRUCTION AND INFERENCE FOR PENME

The projection network is similar to the feed-forward layers in a transformer as it contains two layers with ReLU activation in between with the addition of the Batch Normalization layer, a common element in contrastive learning. The network is trained via contrastive learning which requires a dataset based on a pair of inputs with positive and negative labels. The algorithm 1 data construction process.

---

**Algorithm 1** Psuedo Code Data Construction Projector Network

---
1: **Input** $num\_overall\_negative$            ▷ edit pairing with neighbours of other edits *(Optional hyperparameter)*
2: **Input** $threshold\_edit\_pairings$            ▷ edit-to-edit pairings *(hyperparameter)*
3: **Input** memory = {}            ▷ memory storage
4: **Input** dataset pairs = []            ▷ dataset for training projector network
5: **Input** $Cos(.,.)$            ▷ Cosine Sim function
6: **Input** dataset rows $r_i = [(x_i, y_i, [p_{i1}...p_{ij}], [nb_{i1}...nb_{ij}]), ...]$
7: **for** each $r_i$ element in the dataset **do**
8:     **for** each $p_{ij}$ and $nb_{ij}$ element in the $r_i$ **do**
9:         **dataset pairs** $\leftarrow$ positive pairs$(x_i, p_{ij})$
10:         **dataset pairs** $\leftarrow$ negative pairs$(x_i, nb_{ij})$
11:     **end for**
12:     **for** each $r_t$ in the dataset, where $i \neq t$ **do**
13:         **if** $Cos(x_i, x_t) > threshold$ **then**
14:             **dataset pairs** $\leftarrow$ negative pairs$(x_i, x_t)$
15:         **end if**
16:         **for** each $nb_{ij}$ element in $r_t$ **do**
17:             **memory[i]** $(\leftarrow \cos \text{sim}(x_i, nb_{tj})$, negative pairs $(x_i, nb_{tj}))$
18:         **end for**
19:     **end for**
20: **end for**
21: store $\leftarrow Sort(memory)$            ▷ in descending order
22: **dataset pairs** $\leftarrow memory[0 : \text{negative overall samples}](x_i, n_{tj})$
23: **return** dataset pairs

---

**Algorithm 2** Inference for LLM with PENME

---
1: **Input** $M_l(.)$            ▷ LLM model output at layer l
2: **Input** $M_{proj}(.)$            ▷ Projector network
3: **Input** $D(.,.)$            ▷ Euclidean Distance function
4: **Input** memory = {(keys(K)=vectors, values(V)=(threshold, output))}
5: **Input** $x_t$ user prompt
6: y,$h_l \leftarrow M_l(x_t)$
7: $act_x \leftarrow M_{proj}(h_l)$
8: $selectedKey \leftarrow min_i(D(act_x, K_i))$   ▷ compute Euclidean distance between $act_x$ and keys in memory and extract closet key
9: **if** $D(act_x, selectedKey) < V[selectedKey][threshold]$ **then**
10:     **return** $V[selectedKey][output]$
11: **end if**
12: **return** y

---

## B    PARAPHRASES AND NEIGHBOURS DISTANCE ANALYSIS

Table 2 shows the distance between edits and their respective paraphrases and neighbours across various measurement metrics. From the distances the average paraphrase distance (AvgPD) and average distances between training and test paraphrases (AvgDTTP), we can see that they are generally a little farther than the test paraphrases and are on average a bit farther from the edit than train paraphrases. On the other hand, the average neighbour distance (AvgPN) and average distances between

| Model | Measurement Metric | Training Set | Test Set |
|---|---|---|---|
| | AvgPD | 0.240 | 0.254 |
| | MinPD | 0.0 | 0.02 |
| | MaxPD | 0.829 | 1.59 |
| | AvgND | 1.436 | 1.379 |
| | MinND | 0.803 | 0.616 |
| | MaxND | 1.884 | 1.853 |
| Llama-2-7b | AvgCPFN | 0.348 | 0.893 |
| | | **Training Set vs Test Set** | |
| | AvgDTTP | 0.013 | |
| | MaxDTTP | 1.459 | |
| | MinDTTP | -0.634 | |
| | AvgDTTN | -0.227 | |
| | MaxDTTN | -1.130 | |
| | MinDTTN | 0.0 | |
| | AvgPD | 0.409 | 0.491 |
| | MinPD | 0.0 | 0.002 |
| | MaxPD | 1.375 | 1.381 |
| T5-small | AvgND | 0.468 | 0.534 |
| | MinND | 0.005 | 0.010 |
| | MaxND | 1.384 | 1.386 |
| | AvgCPFN | 0.193 | 0.238 |
| | | **Training Set vs Test Set** | |
| | AvgDTTP | 0.018 | |
| | MaxDTTP | 1.273 | |
| | MinDTTP | -1.290 | |
| | AvgDTTN | -0.276 | |
| | MaxDTTN | -1.341 | |
| | MinDTTN | 0.0 | |
| | AvgPD | 0.378 | 0.349 |
| | MinPD | 0.0 | 0.01 |
| | MaxPD | 1.49 | 1.395 |
| GPT2-XL | AvgND | 1.174 | 1.092 |
| | MinND | 0.227 | 0.368 |
| | MaxND | 1.709 | 1.728 |
| | AvgCPFN | 0.382 | 0.700 |
| | | **Training Set vs Test Set** | |
| | AvgDTTP | 0.008 | |
| | MaxDTTP | 1.368 | |
| | MinDTTP | -1.046 | |
| | AvgDTTN | -0.148 | |
| | MaxDTTN | -0.856 | |
| | MinDTTN | 0.0 | |

Table 2: Distance analysis of distances between edit and its respective paraphrase and neighbours. The metrics for measurement include average/max/min paraphrase distance (AvgPD)(MaxPD)(MinPD), average/max/min neighbour distance (AvgND),(MaxND)(MinND), average/max/min distances between training and test paraphrase (AvgDTTP)(MaxDTTP)(MinDTTP), the average distance between farthest edit and closest neighbour (AvgCPFN) and average/max/min distances between training and test neighbours (AvgDTTN)(MaxDTTN)(MinDTTN)

training and test neighbours (AvgDTTN) show that the test neighbours are a little closer to the edit as compared to the train neighbours.

## C  COMPARISON SCOPING MECHANISM: PENME VERSUS MELO AND GRACE

To demonstrate the improvement in inference time for selecting the appropriate key, we compare PENME with MELO across various sample sizes of edits, ranging from 50 to 300 in increments of 50 shown in table 3. The results show that PENME outperforms MELO in terms of speed and also highlight the number of keys forgotten during training due to the design of its scoping mechanism, as well as the number of entries for which the radius had to be reduced.

| Number of Edits | PENME | MELO/GRACE | | | |
|---|---|---|---|---|---|
| | Runtime (ms) | Runtime (ms) | Codebook Entries | Edits Forgotten | Edit Conflict |
| 50 | $0.024 \pm 0.003$ | $0.316 \pm 0.090$ | 269 | 24 | 21 |
| 100 | $0.115 \pm 0.129$ | $0.364 \pm 0.050$ | 523 | 77 | 66 |
| 150 | $0.188 \pm 0.182$ | $0.624 \pm 0.082$ | 785 | 132 | 114 |
| 200 | $0.279 \pm 0.170$ | $1.423 \pm 0.180$ | 1048 | 188 | 169 |
| 250 | $0.404 \pm 0.170$ | $1.681 \pm 0.205$ | 1319 | 254 | 217 |
| 300 | $0.418 \pm 0.125$ | $2.149 \pm 1.069$ | 1554 | 301 | 268 |

Table 3: Runtime Performance Comparison of PENME versus MELO. For PENME the number of Codebook entries is the same as the number of edits.

# D EXPERIMENTATION AND IMPLEMENTATION DETAILS

## D.1 EXPERIMENTATION SETUP

For our comparative analysis, we contrast against baseline methods such as simple fine-tuning (FT), alongside advanced approaches drawn from relevant literature. These encompass GRACE (Hartvigsen et al., 2024; Yu et al., 2024), employing adapter-based editing with a similarity-based scoping mechanism. SERAC (Mitchell et al., 2022), a multimodal editing approach incorporating a scoping classifier, memory database, and counterfactual model alongside the target model and MEMIT (Meng et al., 2022b) an editing approach designed for decoder only model adopts a model-editing strategy by identifying and updating knowledge-contained model layers' weight matrices. Its

In evaluating our approach, we adhere to metrics outlined in section 3. Regarding generalization, we define a paraphrase as generalized if it aligns with the correct edit and falls below its distance threshold. For assessing locality, we maintain that locality is preserved when the distance between matched edits exceeds its threshold. Any other instances are categorized as misfires. It is important to note that (Hartvigsen et al., 2024; Yu et al., 2024) utilize token F1 Accuracy and (Mitchell et al., 2022) use a metric based on token probabilities. These metrics are softer in nature which allows for higher scores.

### D.1.1 COMPUTATION RESOURCES

Training for all projector networks is conducted on NVIDIA P100 GPU with 16GB VRAM. A larger VRAM or RAM capacity is only necessary for the initial extraction of layer representations from the pre-trained language models. For the evaluation of approaches from relevant literature, some of which demanded greater computational resources, we employed NVIDIA A100 GPU with 40GB, and 80GB VRAM. All editing approaches where supported are implemented using the default configurations provided in the Easy-Editor library (Wang et al., 2023a). It is important to note that not all models are supported across all editing methods. For instance Llama-2-7b is not supported for MELO. For some models such as T5-small limited support is provided therefore we utilize the code provided by the papers authors.

The inference pipeline for PENME is given in 2.

### D.1.2 HYPERPARAMETERS

For training projector networks we utilize the Adam optimizer. we experiment with various learning rates $1e^{1-2}, 2e^{1-2}, 3e^{1-2}$. we find that a moderate learning rate is required to learn faster while not overfitting, hence we choose $1e^{1-2}$, with a learning rate decay rate of $0.01$. All projection networks are trained for 200 epochs using a batch size of 8192 and an early stopping patience of 8 epochs. For selecting the margin $m$ in the contrastive learning cost function we ablate on the hyperparameter m for the GPT2-XL model. The table 4 shows the margin m along with the adjustment to $\tau$ for balanced results for generalization and locality. It can be observed from the table to achieve high-performance minimum value of 30 needs to be utilized. The higher the the value for $m$ the better the score for localization. The value chosen is 40 which has the most balanced results.

| Margin $m$ | Threshold Adjustment $\tau$ | Generalization | Locality |
|---|---|---|---|
| 10 | 0 | 0.634 | 0.831 |
| 20 | 3 | 0.891 | 0.880 |
| 30 | 6 | 0.958 | 0.948 |
| 40 | 8 | 0.967 | 0.977 |
| 50 | 11 | **0.978** | 0.965 |
| 60 | 13 | 0.976 | 0.986 |
| 70 | 17 | 0.973 | 0.976 |
| 80 | 17 | 0.973 | 0.976 |
| 90 | 20 | 0.928 | **0.986** |

Table 4: The table shows how the performance changes along with the required threshold adjustment to $\tau$ as margin $m$ in contrastive loss is changed

| | | | ZsRE | | | Counterfact | | | |
|---|---|---|---|---|---|---|---|---|---|
| Metric | Pair Type | Score | Precision | Recall | F1 | Value | Precision | Recall | F1 |
| Jaccard Similarity | $(x_i, p_{ij})$ | 0.399 | - | - | - | 0.401 | - | - | - |
| Jaccard Similarity | $(x_i, nb_{ij})$ | 0.086 | - | - | - | 0.430 | - | - | - |
| ROUGE-1 | $(x_i, p_{ij})$ | - | 0.321 | 0.315 | 0.316 | - | 0.310 | 0.325 | 0.307 |
| ROUGE-1 | $(x_i, nb_{ij})$ | - | 0.076 | 0.087 | 0.079 | - | 0.295 | 0.293 | 0.290 |
| ROUGE-2 | $(x_i, p_{ij})$ | - | 0.189 | 0.194 | 0.194 | - | 0.189 | 0.198 | 0.184 |
| ROUGE-2 | $(x_i, nb_{ij})$ | - | 0.008 | 0.008 | 0.008 | - | 0.205 | 0.203 | 0.201 |
| ROUGE-L | $(x_i, p_{ij})$ | - | 0.299 | 0.294 | 0.293 | - | 0.299 | 0.312 | 0.295 |
| ROUGE-L | $(x_i, nb_{ij})$ | - | 0.070 | 0.080 | 0.073 | - | 0.294 | 0.292 | 0.289 |

Table 5: Comparison between ZsRE and Counterfact for token overlap metrics

### D.2 DATA PROCESSING

**Counterfact:** Each row in the Counterfact consists of an edit prompt, two paraphrase prompts, multiple neighbourhood prompts and an edit label $x_i, y_i, [p_1, p_2], [nb_{i1}...nb_{ij}])$. For the training dataset, we extract the edit prompt $x_i$, one randomly sampled paraphrase $p_i$ and half the neighbourhood prompts $nb_{ij}$. For creating additional paraphrases for the training set we utilize the extracted edit prompt and paraphrase prompt as input to ChatGPT and use it to generate three additional paraphrases for training. We ensure that the generated paraphrase follows the $(s, r, o^*)$ triplet format that the dataset uses. The test set for locality and generalization compromises of the paraphrase and neighbours not sampled from the training set.

**zsRE:** The zsRE dataset comprises of rows containing a sample question, its corresponding new label, and multiple rephrased questions along with its filtered rephrased questions. We constructed this dataset following methodologies established in the relevant literature. A balanced subset of paraphrases are derived from the filtered rephrased questions for training and testing purposes. For neighbouring samples, we randomly selected an equal number of questions from the NQ dataset for training and testing while ensuring no overlap in questions.

To highlight the lexicality issue in the datasets, we compute several token overlap metrics between pairs of (edits, paraphrases) $(x_i, p_{ij})$ and (edits, neighbors) $(x_i, nb_{ij})$, and present text examples from both datasets in the table 5 and 6. From the token overlap metrics table, it is evident that the edit prompt and neighbors show high overlap in Counterfact, whereas the overlap is minimal in ZsRE. This, coupled with the experiment in section §6.1, highlights the significant challenges observed in the Counterfact dataset.

| Counterfact | | | ZsRE | | |
|---|---|---|---|---|---|
| **Edit** | **Paraphrase** | **Neighbour** | **Edit** | **Paraphrase** | **Neighbour NQ dataset** |
| The twin city of Cologne is | What is the twin city of Cologne? It is | The twin city of London is | Which river system contains Laborec? | What river system does Laborec contain? | Where does the last name serrano come from? |
| Alexander Zinoviev works in the area of | Alexander Zinoviev's domain of work is | TFred W. Riggs works in the area of | Which airport does Air Seychelles operate in? | Which airport is closely linked to Air Seychelles? | How many students attend chippewa valley high school? |
| The original language of Kondura was | The language of Kondura is | The original language of Water was | The country of origin for Kala Pul is what? | Which was the country for Kala Pul? | "When do the new sky sports channels launch? |
| Thomas Arne died in the city of | Thomas Arne lost their life at | Bill Brandt died in the city of | What label was responsible for Wild World? | What was the label Wild World? | Who composed the music for avengers infinity war? |

Table 6: Random samples from the Counterfact and ZsRE datasets.

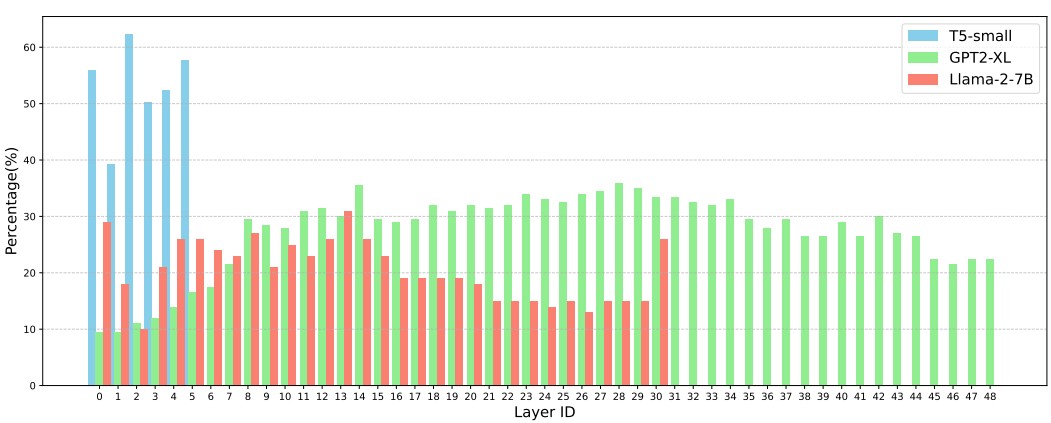

Figure 7: Percentage of samples *where edits are closer* to neighbours as compared to paraphrases in the representations space of different models across all layers. T5-small, GPT2-XL and Llama-2-7b have 6, 32, 48 layers respectively.

# E    PROJECTOR NETWORK AND LEXICAL DOMINANCE

## E.1    LEXICAL DOMINANCE LAYER ANALYSIS

Figure 7 shows the percentage of edits samples where neighbours were closer to the edits for all models across all layers.

## E.2    LAYER-WISE ANALYSIS OF THE PROJECTOR NETWORK

Figure 8 shows how the results for generalization and locality for the T5-small model. The results suggest that performance remains largely consistent; however, training tends to require more time to converge at higher layers.

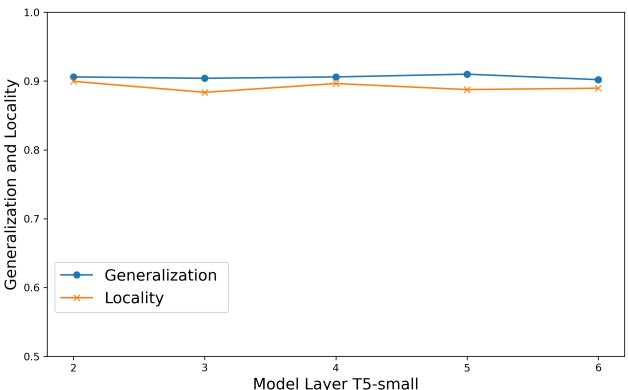

Figure 8: Generalization and locality scores for various projector networks trained on layers of T5-small using 500 samples from Counterfact.

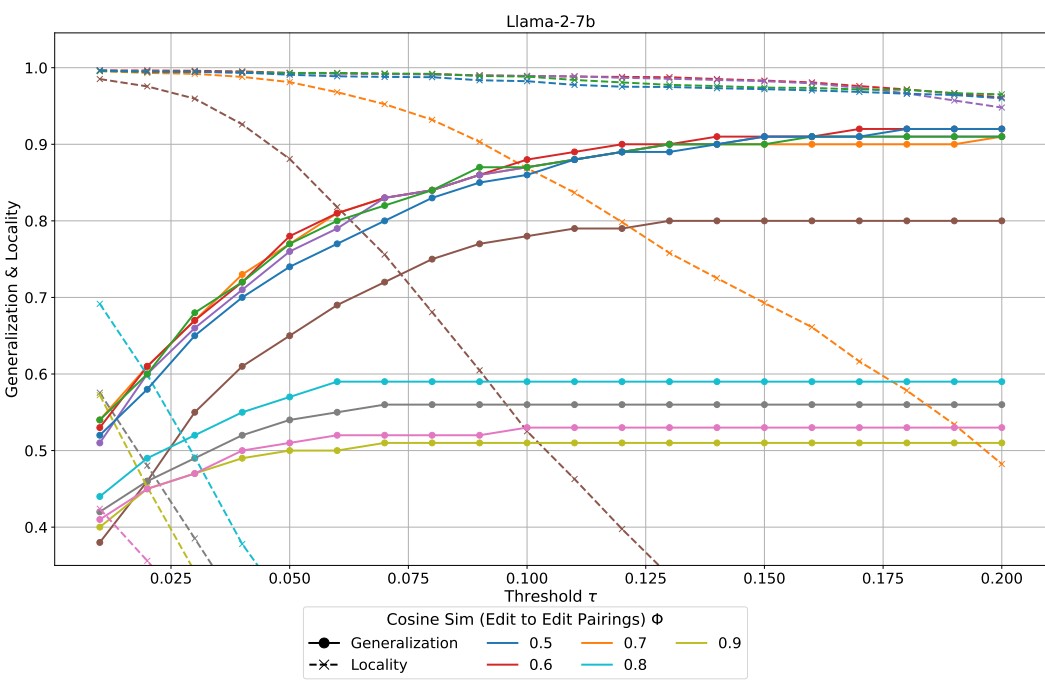

Figure 9: Generalization and Locality trade-off a function of varying distance thresholds $\tau$ and $\phi$

# F VISUALIZATIONS

## F.1 GENERALIZATION AND LOCALITY LLAMA-2-7B

9 shows generalization and locality trade-off a function of varying distance thresholds $\tau$ and $\phi$ for Llama-2-7b model.

## F.2 PCA

Figures 10 and 11 present the two-dimensional PCA of the model representations and projector network representations for the Llama-2-7b and GPT2-XL models, respectively. The visual demonstrates that neighboring prompts are closely aligned with edit prompts, while edit prompts also show

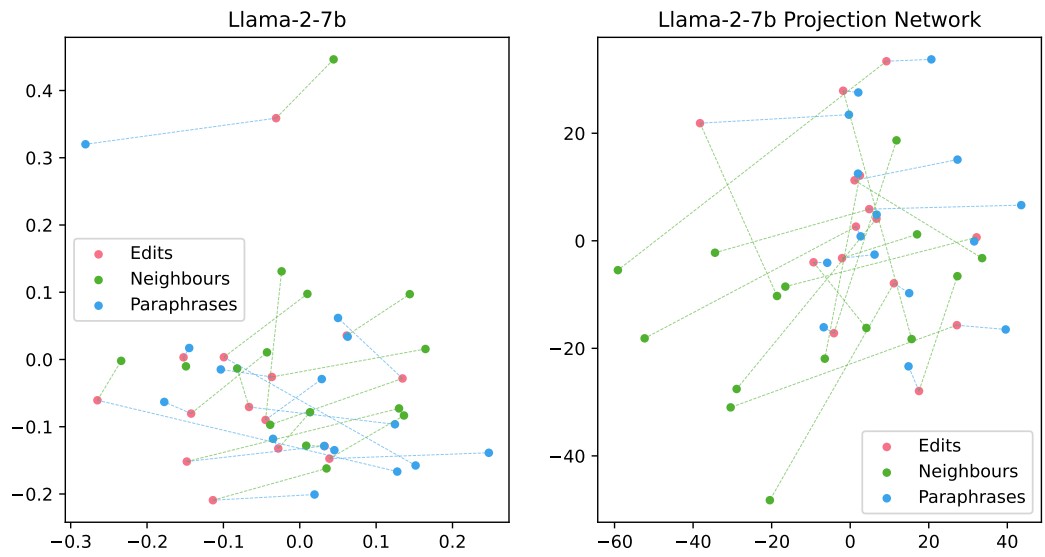

Figure 10: Generalization and locality scores for various project or networks trained on layers of T5-small using 500 samples from Counterfact.

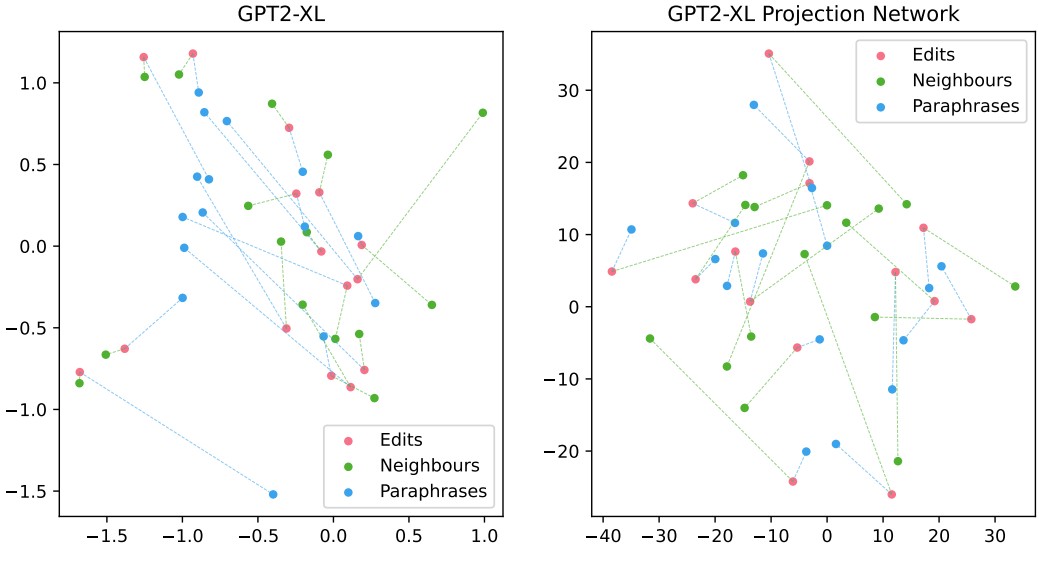

Figure 11: Two dimensional PCA on GPT2-XL model representation and the trained projejctor network.

proximity to other edit prompts within the original model representations. The projector network, however, effectively mitigates this effect by learning a disentangled representation space.

## G    ERROR ANALYSIS PROJECTOR NETWORK

To investigate the reasons behind failures in PENME, we performed a comprehensive error analysis across our models. Our findings indicate that contrastive learning significantly mitigates lexical

dominance. However, due to the inherent variability in lexical pattern distribution within the dataset, there remains potential for further optimization in the projection phase.

The training process of the projector network does not lead to uniform distances between each edit, its paraphrases and neighbours for all samples. This paired with individually varying thresholds for edits leads to misfires. To illustrate this problem, we format the results of each dataset sample for automatic inspection. For all paraphrases and neighbours in the test set, we extract the nearest key/edit, the ground truth edit/key, the distance to the nearest key/edit, and the distance to the ground truth edit/key. Table 7 shows rouge scores (Lin, 2004) for two possible scenarios i.e. success and failure of generalization and locality. We also show separately the score for where generalization failure occurs due to distance not meeting the set threshold. Moreover, since failures can occur in similarities with unrelated edits we show locality and paraphrase failure with both ground truth edit and matched edit.

For cases of successful generalization, we observe a substantial uni-gram overlap and a moderate bi-gram overlap between the edited sentences and their paraphrases. The ROUGE-L scores are similarly high for these metrics, indicating that the sentences likely share similar tokens in the same sequence. This implies that the attention mechanism produces similar representations, leading to a high degree of similarity. For locality success, we can see that although there is significant token overlap between neighbours and their target edits, the neighbours had higher similarity with some other edits with low token overlap, this means our approach of pushing neighbouring sentences farther away is able to generalize to unseen neighbours.

In cases of generalization failure, the ROUGE scores for paraphrases compared with the ground truth are slightly lower than those observed in successful instances. Although there is some token overlap with the target edits, the matched edits exhibit even less token overlap. On the other hand for locality failure, we can see that the prediction case token overlap is higher as compared to locality success, moreover, the overlap is higher as compared to ground truth edits. Thus lexicality based similarity is not the issue but rather the varying thresholds, which in some cases are large leads to misfires.

## H  LIMITATIONS

Training the projection network in PENME using the contrastive learning scheme is sensitive, requiring tuning of hyperparameters such as the learning rate and contrastive loss margin. Effective network training also hinges on the careful construction of training data, which requires careful consideration of the number of edit pairings with other dataset neighbours and edit-to-edit pairings. Finally, the thresholds for the memory-based retrieval system, though dynamically determined from training data, can vary across different models, necessitating adjustments to the alpha ($a$) parameter for each model.

| Model | Rouge-1 2.5% CI | Rouge-1 97% CI | Rouge-2 2.5% CI | Rouge-2 97% CI | RougeL 2.5% CI | RougeL 97% CI |
|---|---|---|---|---|---|---|
| | | | **Generalization Success** | | | |
| T5-small | 1.00 | 0.95 | 1.05 | 0.706 | 0.65 | 0.75 |
| Llama-2-7b | 0.629 | 0.639 | 0.382 | 0.394 | 0.608 | 0.619 |
| GPT2-XL | 0.655 | 0.666 | 0.403 | 0.417 | 0.642 | 0.653 |
| | | | **Generalization Failure (prediction)** | | | |
| T5-small | 1.00 | 0.95 | 1.05 | 0.706 | 0.65 | 0.75 |
| Llama-2-7b | 0.133 | 0.173 | 0.056 | 0.091 | 0.125 | 0.162 |
| GPT2-XL | 0.122616 | 0.160 | 0.056 | 0.090 | 0.117 | 0.153 |
| | | | **Generalization Failure (ground truth)** | | | |
| T5-small | 1.00 | 0.95 | 1.05 | 0.706 | 0.65 | 0.75 |
| Llama-2-7b | 0.488 | 0.518 | 0.270 | 0.296 | 0.460 | 0.489 |
| GPT2-XL | 0.501 | 0.527 | 0.284 | 0.310 | 0.474 | 0.500 |
| | | | **Locality Success (prediction)** | | | |
| T5-small | 0.100 | 0.104 | 0.011 | 0.013 | 0.096 | 0.099 |
| Llama-2-7b | 0.100 | 0.104 | 0.011 | 0.013 | 0.096 | 0.099 |
| GPT2-XL | 0.095 | 0.100 | 0.011 | 0.013 | 0.092 | 0.095 |
| | | | **Locality Success (ground truth)** | | | |
| T5-small | 0.100 | 0.104 | 0.011 | 0.013 | 0.096 | 0.099 |
| Llama-2-7b | 0.487 | 0.518 | 0.269 | 0.296 | 0.459 | 0.489 |
| GPT2-XL | 0.176 | 0.217 | 0.036 | 0.059 | 0.173 | 0.211 |
| | | | **Locality Failure (prediction)** | | | |
| T5-small | 0.566 | 0.577 | 0.390 | 0.403 | 0.562 | 0.574 |
| Llama-2-7b | 0.259 | 0.277 | 0.148 | 0.164 | 0.247 | 0.264 |
| GPT2-XL | 0.254 | 0.273 | 0.147 | 0.164 | 0.244 | 0.262 |
| | | | **Locality Failure (ground truth)** | | | |
| T5-small | 0.203 | 0.212 | 0.052 | 0.058 | 0.197 | 0.206 |
| Llama-2-7b | 0.201 | 0.206 | 0.049 | 0.053 | 0.195 | 0.201 |
| GPT2-XL | 0.207 | 0.218 | 0.052 | 0.059 | 0.201 | 0.212 |
| | | | **Generalization Distance Failure** | | | |
| T5-small | 1.00 | 0.95 | 1.05 | 0.706 | 0.65 | 0.75 |
| GPT2-XL | 0.522 | 0.551 | 0.279 | 0.309 | 0.484 | 0.512 |
| Llama-2-7b | 0.495 | 0.579 | 0.252 | 0.324 | 0.455 | 0.529 |

Table 7: ROUGE Evaluation Scores