# OpenReview forum: "Resolving Lexical Bias in Edit Scoping with Projector Editor Networks"
_ICLR.cc/2025/Conference — Submitted to ICLR 2025_

### Official Review · Reviewer_uwxU · 2024-10-27

**Soundness:** 3
**Presentation:** 2
**Contribution:** 2
**Rating:** 5
**Confidence:** 3

**Summary:**

The paper proposes to address lexical bias in continual model editing (i.e., token similarity affecting edit decisions). The framework is similar to existing clustering-based setups (e.g., GRACE [1]). This seems to be achieved by explicitly training a projection network and discouraging exploiting lexical correlations. The paper shows improved performance on CounterFact and zsRE.

[1] Aging with GRACE: Lifelong Model Editing with Discrete Key-Value Adaptors (Hartvigsen et al., 2023)

**Strengths:**

The paper highlights the problem of lexical bias in clustering-based editing approaches, which can raise awareness of this particular issue.

**Weaknesses:**

It is difficult to tell exactly what parts of the paper are novel contributions. I think the main difference is that in GRACE the cookbook representations are manually maintained whereas here they're learned. The related work section needs to tell the reader why this work is different from the previous works, not just describing them.

**Questions:**

N/A

---

> ### Author Response · Authors · 2024-11-20
> **Response**
>
> We thank Reviewer uwxU for their time and feedback, which we address below.
>
> **1. It is difficult to tell exactly what parts of the paper are novel contributions. I think the main difference is that in GRACE the cookbook representations are manually maintained whereas here they're learned. The related work section needs to tell the reader why this work is different from the previous works, not just describing them.**
>
> **Novelty:** This is the first work that identifies the issue of lexical bias in model editing. The explicit modelling of a representation space designed to improve paraphrase success while minimizing mismatch with neighbouring examples is a novel solution to the problem at hand. To the best of our knowledge, PENME is the only model editing method designed to maintain both high locality and generalization. The results in Table 1 demonstrate the efficacy of PENME where none of the other methods, whether weight-preserving or weight-modifying, improves both locality and generalization. We have highlighted the novelty of our work in the introduction. We have further improved the Related Work section with a more explicit comparison between our method and other model-editing methods.
>
> The **key distinction between GRACE [1] and PENME** lies in their approach to model representations: GRACE utilizes a codebook with cached representations as keys, whereas PENME employs learned representations generated by a projection network. GRACE adopts small thresholds or deferral radii to address challenges with neighbourhood prompts; however, this approach is less effective when lexically similar prompts are in close proximity. Additionally, maintaining small radii necessitates storing paraphrases as extra codebook entries alongside their corresponding edits to enhance generalization, which significantly increases retrieval time. The radius-based design further introduces the risk of edit forgetting, as overlapping cluster radii for similar edits necessitate the resizing of radii. This creates a trade-off between locality and generalization, where small radii preserve locality at the expense of generalization, and larger radii enhance generalization but reduce edit localization. In contrast, PENME effectively addresses these issues by managing lexically similar prompts more robustly and leveraging its learned representation space to restrict each edit to a single codebook entry, thereby simplifying storage and retrieval processes.
>
> To illustrate the aforementioned challenges in GRACE and how PENME alleviates these issues, we present an ablation experiment comparing PENME with GRACE across various sample sizes of edits, ranging from 50 to 300 in increments of 50, on the Counterfact dataset, as shown in the table below. The results indicate that PENME outperforms  GRACE in speed due to the number of codebook entries, while also revealing key shortcomings in their scoping methods. Specifically, the 'Edits Forgotten' and 'Edit Conflict' columns highlight the significant number of keys lost during training due to conflicts in the deferral radii of edits.
>
> | **Number of Edits** | **PENME**         | **MELO/GRACE**            |                          |                          |                          |
> |---------------------|-------------------|---------------------------|--------------------------|--------------------------|--------------------------|
> |                     | **Runtime (ms)**  | **Runtime (ms)**          | **Codebook Entries**     | **Edits Forgotten**      | **Edit Conflict**        |
> |---------------------|-------------------|---------------------------|--------------------------|--------------------------|--------------------------|
> | 50                  | 0.024 ± 0.003     | 0.316 ± 0.090             | 269                      | 24                       | 21                       |
> | 100                 | 0.115 ± 0.129     | 0.364 ± 0.050             | 523                      | 77                       | 66                       |
> | 150                 | 0.188 ± 0.182     | 0.624 ± 0.082             | 785                      | 132                      | 114                      |
> | 200                 | 0.279 ± 0.170     | 1.423 ± 0.180             | 1048                     | 188                      | 169                      |
> | 250                 | 0.404 ± 0.170     | 1.681 ± 0.205             | 1319                     | 254                      | 217                      |
> | 300                 | 0.418 ± 0.125     | 2.149 ± 1.069             | 1554                     | 301                      | 268                      |
>
> We have updated Section (*RELATED WORK*) to highlight this better.

---

> > ### Author Response · Authors · 2024-11-25
> > **Official Comment Authors**
> >
> > We recognize that the primary concern of the reviewer was about what makes our work novel, as well as its differences from previous related work. We hope that our previous rebuttal was able to show exactly what our novel contributions are and how they relate to previous work. We thank the reviewer for raising this point and have added clarity to our revisions.
> >
> > Please let us know before the end of the rebuttal period if there is anything further we can clarify.

---

### Official Review · Reviewer_hiiz · 2024-10-28

**Soundness:** 3
**Presentation:** 3
**Contribution:** 3
**Rating:** 6
**Confidence:** 4

**Summary:**

The paper introduces Projector Editor Networks for Model Editing (PENME), a novel approach to improving large language model editing techniques. PENME addresses the wrong when deal with incorrect edits on irrelevant prompts with similar words by using contrastive learning to create an optimized representation space. This space allows precise localization of edits by maintaining distance between irrelevant prompts while keeping paraphrases close. The empirical study demonstrates that PENME achieves great results in model editing.

**Strengths:**

* Propose the lexical bias in Model editing which is a new aspect to improve the performance of model editing.
* Propose a projection network that maps the model’s representation space to a new representation space where lexical dominance is minimized

**Weaknesses:**

* This paper suggests that lexical bias refers to different editing subjects with the same relation, such as "The twin city of Pittsburgh is" and "The twin city of Portsmouth is." However, the prevalence of such cases in the CounterFact and ZsRE datasets is unclear.
﻿
* Figure 3 and Figure 7 illustrate the "Percentage of samples where edits are closer to unrelated neighbors," but this is insufficient to demonstrate lexical bias. At lower model layers, high similarity may result from underdeveloped sentence representations, while at higher layers, the reduced percentage indicates greater differentiation between sentences.
﻿
* The results in Table 1 show that GRACE is a strong baseline. PENME, which extends GRACE by using a projection network to map data representations, needs to clearly highlight the differences between PENME and GRACE.
﻿
* PENME focuses on addressing lexical bias, so it should perform well on Loc and Para. However, in Table 1, only Para shows improvement, which is insufficient to fully support the paper's contributions.

**Questions:**

* It is better to give more example to shown what is  lexical bias and lexical overlap in the paper.
* Some results in Table 1 is not clear enough (close to 0.0), such as why GRACE on zsRE get the 0.00 in Loc on Llama2-7b?
* In Figure 6, it is better to add the resutls on GRACE.

---

> ### Author Response · Authors · 2024-11-20
> **Part 1**
>
> We appreciate Reviewer hiiz's time and attention in reviewing our paper, as well as the detailed feedback, which we intend to address below.
>
> **1. This paper suggests that lexical bias refers to different editing subjects with the same relation, such as "The twin city of Pittsburgh is" and "The twin city of Portsmouth is." However, the prevalence of such cases in the CounterFact and ZsRE datasets is unclear.**
>
> **It is better to give more example to shown what is lexical bias and lexical overlap in the paper.**
>
> To quantify lexical bias, we compute token overlap using Jaccard similarity and ROUGE metrics between pairs of (edits, paraphrases) $(x_i, p_{ij})$ and (edits, neighbours) $(x_i, nb_{ij})$, and also present examples from both datasets in the tables below. From the token overlap metrics table, it is evident that the edit prompt and neighbours show a high overlap in Counterfact, whereas the overlap is minimal in ZsRE. The table with dataset examples further demonstrates that neighbours share a similar lexical structure but often refer to different entities. Interestingly, these entities can also be semantically related, as seen in the fourth example, where Thomas Arne and Bill Brandt are both well-known British figures.
>
> This, coupled with the experiment in  Section 6.1 (*LEXICAL DOMINANCE*) highlights the challenging nature of the Counterfact dataset, attributable to its inherent lexical bias.
>
> | **ZsRE**            |                     |           |               |            |        |           |  **Counterfact**             |            |        |
> |---------------------|---------------------|-----------|---------------|------------|--------|-----------|---------------|------------|--------|
> | **Metric**          | **Pair Type**       | **Score** | **Precision** | **Recall** | **F1** | **Value** | **Precision** | **Recall** | **F1** |
> | Jaccard Similarity  | $(x_i,p_{ij})$       | 0.399     | -             | -          | -      | 0.401     | -             | -          | -      |
> | Jaccard Similarity  | $(x_i,nb_{ij})$      | 0.086     | -             | -          | -      | 0.430     | -             | -          | -      |
> | ROUGE-1             | $(x_i,p_{ij})$       | -         | 0.321         | 0.315      | 0.316  | -         | 0.310         | 0.325      | 0.307  |
> | ROUGE-1             | $(x_i,nb_{ij})$      | -         | 0.076         | 0.087      | 0.079  | -         | 0.295         | 0.293      | 0.290  |
> | ROUGE-2             | $(x_i,p_{ij})$       | -         | 0.189         | 0.194      | 0.194  | -         | 0.189         | 0.198      | 0.184  |
> | ROUGE-2             | $(x_i,nb_{ij})$      | -         | 0.008         | 0.008      | 0.008  | -         | 0.205         | 0.203      | 0.201  |
> | ROUGE-L             | $(x_i,p_{ij})$       | -         | 0.299         | 0.294      | 0.293  | -         | 0.299         | 0.312      | 0.295  |
> | ROUGE-L             | $(x_i,nb_{ij})$      | -         | 0.070         | 0.080      | 0.073  | -         | 0.294         | 0.292      | 0.289  |
>
> ---
>
> *Random Samples from the Counterfact and ZsRE Datasets*
>
> | **Counterfact**                          | **Edit**                                    | **Paraphrase**                               | **Neighbour**                          |
> |------------------------------------------|---------------------------------------------|---------------------------------------------|----------------------------------------|
> | |The twin city of Cologne is | What is the twin city of Cologne? It is     | The twin city of London is ||
> | |Alexander Zinoviev works in the area of  | Alexander Zinoviev's domain of work is      | Fred W. Riggs works in the area of  |                                        |
> | |The original language of Kondura was     | The language of Kondura is     | The language of Taal is                     |                                        |
> | |Thomas Arne died in the city of   | Thomas Arne lost their life at      | Bill Brandt died in the city of   |
>  |
>
>
> | **ZsRE**                          | **Edit**                                    | **Paraphrase**                               | **Neighbour**                          |
> |------------------------------------------|---------------------------------------------|---------------------------------------------|----------------------------------------|
> | |Which river system contains Laborec?     | What river system does Laborec contain? | Where does the last name Serrano come from? |
> | |Which airport does Air Seychelles operate in? | Which airport is closely linked to Air Seychelles? | How many students attend Chippewa Valley High School? |
> | |The country of origin for Kala Pul is what? | Which was the country for Kala Pul? | When do the new Sky Sports channels launch? |
> | |What label was responsible for Wild World? | What was the label Wild World? | Who composed the music for Avengers: Infinity War? |
>
> ---

---

> > ### Comment · Reviewer_hiiz · 2024-11-21
> > **Results on Neighbour data (Loc)**
> >
> > The author in Part 1 of the analysis notes that "it is evident that the edit prompt and neighbours show a high overlap in Counterfact, whereas the overlap is minimal in ZsRE." The bias the author aims to address concerns data with similar structures but different entities.
> >
> >
> > According to your analysis, such data is more prevalent among the neighbours. Therefore, PENME should seemingly perform better on Loc, especially on the Counterfact dataset. However, this contradicts the results in Table 1, where GRACE performs better, leaving the reasons for the overall performance improvement unexplained.

---

> ### Author Response · Authors · 2024-11-20
> **Part 2**
>
> **2. The results in Table 1 show that GRACE is a strong baseline. PENME, which extends GRACE by using a projection network to map data representations, needs to clearly highlight the differences between PENME and GRACE.**
>
> Please refer to our central response for clarification.
>
> **3. Figure 3 and Figure 7 illustrate the "Percentage of samples where edits are closer to unrelated neighbors," but this is insufficient to demonstrate lexical bias. At lower model layers, high similarity may result from underdeveloped sentence representations, while at higher layers, the reduced percentage indicates greater differentiation between sentences.**
>
> We agree that the lower layers of a model may have underdeveloped sentence representations, while higher layers may have developed a better sense of semantic similarity. The Figure 3 bar chart shows that the lexical bias exists in higher layers as well; however, it is relatively low compared to lower layers. Moreover, the issue of lexical bias has also been observed in the recent NeurIPs paper [1], which specifically analyzed the last layer representations of a diverse set of large language models.
>
> **4. PENME focuses on addressing lexical bias, so it should perform well on Loc and Para. However, in Table 1, only Para shows improvement, which is insufficient to fully support the paper's contributions.**
>
> PENME  achieves a balanced outcome, delivering high scores in both generalization and locality without compromising performance in either aspect. The deferral radius in GRACE [2] and MELO [3] which utilize the same retrieval system is dynamically adjusted, initially set to a small epsilon to avoid interference with other prompts, ensuring high locality, while still permitting the execution of some paraphrases for generalization. It is important to note that multiple paraphrases during training are essential for achieving effective generalization. While the default deferral radius is small, enabling high locality, it leads to a redundant solution with minimal generalization as evident from the results. If the radius is expanded it leads to better generalization but at the cost of locality, given that lexically similar neighbours exhibit proximity to the edits, this leads to low locality scores.
>
> **5. Some results in Table 1 is not clear enough (close to 0.0), such as why GRACE on zsRE get the 0.00 in Loc on Llama2-7b?**
>
> For our evaluations, we employ the default parameters and layer settings provided in the EasyEdit library [4]. It is important to note that Llama-2-7b operates within a representation space that differs significantly from that of the GPT series models. Alternative model editing approaches, such as MEMIT, have been shown to perform sub-optimally for model editing tasks in this context. We direct the authors to a recent NeurIPs paper [5], where the appendix Section D (Appendix /Additional Experiments) demonstrates that the results of MEMIT and GRACE for the LLaMA model align closely with those observed in our study.
>
> **6. In Figure 6, it is better to add the resutls on GRACE.**
>
> Conducting model editing methods evaluations across different scaling experiments is computationally intensive. To address concerns related to scalability, we focused our primary experiments on comparing PENME's performance with other editing approaches for 2,000 edits, as shown in Table 1. Notably, experiments in relevant literature typically evaluate performance with up to 1,000 edits [2], [3], [5].
>
> [1] Sri Harsha Dumpala, Aman Jaiswal, Chandramouli Sastry, Evangelos Milios, Sageev Oore, and
> Hassan Sajjad (2024). Sugarcrepe++ dataset: Vision-language model sensitivity to semantic and lexical alterations. In Proceedings of NeurIPS.
>
> [2] Tom Hartvigsen, Swami Sankaranarayanan, Hamid Palangi, Yoon Kim, and Marzyeh Ghassemi. Aging with grace: Lifelong model editing with discrete key-value adaptors. Advances in Neural Information Processing Systems
>
> [3] Lang Yu, Qin Chen, Jie Zhou, and Liang He. Melo: Enhancing model editing with neuron-indexed dynamic lora. In Proceedings of the AAAI Conference on Artificial Intelligence.
>
> [4] Peng Wang, Ningyu Zhang, Xin Xie, Yunzhi Yao, Bozhong Tian, Mengru Wang, Zekun Xi, Siyuan
> Cheng, Kangwei Liu, Guozhou Zheng, et al. Easyedit: An easy-to-use knowledge editing framework for large language models. arXiv preprint arXiv:2308.07269, 2023a
>
> [5] Xiusheng Huang, Jiaxiang Liu, Yequan Wang, and Kang Liu. Reasons and solutions for the decline in model performance after editing. In Proceedings of NeurIPS.

---

> ### Author Response · Authors · 2024-11-21
> **Response (Results on Neighbour data (Loc))**
>
> As highlighted in our response to Part 1, GRACE falls short of satisfying a fundamental criterion of model editing: meaningful generalization. While its high locality may initially appear noteworthy, the results underscore a critical limitation of GRACE, as it achieves less than $2$\% generalization on Counterfact across all models. This performance is comparable to a simplistic input-matching mechanism reliant on caching. The primary scientific challenge in model editing lies not only in preserving locality but also in ensuring that targeted factual updates propagate effectively across semantically related queries. PENME is designed to address both the challenges of generalization and locality, demonstrating strong performance in both areas.
>
>
> To illustrate how GRACE's performance evolves as its deferral radii are adjusted in an attempt to approach PENME's generalization performance, we increase the deferral radii and evaluate the approach on 2000 Counterfact samples for GPT2-XL, as shown in the table below.
>
> | **PENME**       |             |             |             | **GRACE**      |             |             |
> |------------------|-------------|-------------|-------------|----------------|-------------|-------------|
> | **ES**          | **LOC**     | **PARA**    |             | **ES**         | **LOC**     | **PARA**    |
> | **1.00**       | **0.847**   | **0.875**   |             | **1.00**       | 0.171       | 0.767       |

---

> ### Comment · Reviewer_hiiz · 2024-11-22
> **Training of PROJECTION NETWORK**
>
> Thank you for your response. The results indeed demonstrate the advantages of PENME in scenarios involving extensive continual editing.
>
> However, it seems to rely on a crucial component, the "PROJECTION NETWORK," which raises a question: Is the PROJECTION NETWORK pre-trained, or does it play a role in continual learning?
>
> In the context of continual editing, the assumption is that the model do not have all the knowledge updates at once but receive them sequentially.
>
> Therefore, the training of the PROJECTION NETWORK appears to be a small-sample training problem. Has the author considered this aspect?
>
> You could check the paper "https://arxiv.org/pdf/2405.14768", they didn't use all edits before editing, but for PROJECTION NETWORK, does it need all edits for training?

---

> > ### Author Response · Authors · 2024-11-22
> > **Response (Training of PROJECTION NETWORK)**
> >
> > We appreciate the opportunity to clarify our approach. Our evaluations were conducted within a batch editing framework, and we thank the reviewer for highlighting the problem of continual editing. To address this, we conduct an experiment using Llama-2-7b, leveraging the pretrained projector network from our original experiments. We randomly sample 1,000 unseen instances from ZsRE to evaluate PNEME and compare it to WISE [6]. The results, presented in the table below, show that PNEME maintains strong performance in both locality and generalization in a continual editing setting. In contrast, while WISE emphasizes high locality, it experiences trade-offs in the form of edit forgetting and reduced generalization.
> >
> > Due to current computational resource constraints, we are in the process of compiling the experimental results for all models for this setting. Once finalized, these results will be included in the manuscript.
> >
> > | **PENME**            |            |            |            | **WISE**             |            |            |            |
> > |-----------------------|------------|------------|------------|-----------------------|------------|------------|------------|
> > | **ES**               | **LOC**    | **PARA**   | **Score**  | **ES**               | **LOC**    | **PARA**   | **Score**  |
> > |||||||||
> > | **1.00**             | 0.917      | **0.861**  | **0.93**   | 0.77                 | **1.00**   | 0.72       | 0.83       |
> >
> >
> > [6] Peng Wang, Zexi Li, Ningyu Zhang, Ziwen Xu, Yunzhi Yao, Yong Jiang, Pengjun Xie, Fei Huang, and Huajun Chen. Wise: Rethinking the knowledge memory for lifelong model editing of large
> > language models.

---

> > > ### Comment · Reviewer_hiiz · 2024-11-25
> > > **Conclusion**
> > >
> > > Thanks for your time and effort during the rebuttal. I think you have addressed most of my concern in this paper. So I decide to revise my scores accordingly.

---

### Official Review · Reviewer_UAfd · 2024-10-29

**Soundness:** 3
**Presentation:** 3
**Contribution:** 3
**Rating:** 6
**Confidence:** 2

**Summary:**

This paper addresses an important question in model editing: the tradeoff between generalization (e.g., paraphrase handling) and locality (e.g., avoiding unintended edits on irrelevant queries). To tackle this issue, the authors propose PENME, which consists of two components: (1) a projection network trained with a contrastive objective to separate paraphrased and irrelevant prompts in the representation space, and (2) a memory-based retrieval scheme that enhances editing precision by applying a similarity threshold as a scoping mechanism. Experiments on three models demonstrate the effectiveness of PENME compared to other baselines.

**Strengths:**

1. This paper addresses an important topic in model editing: the tradeoff between generalization and locality. Though the problem is already well-defined, the two proposed methods are simple yet effective to improve editing's effectiveness.
3. The experimental results are significant, showing improvements over several state-of-the-art editing methods.

**Weaknesses:**

1. I don’t have strong negative feedback on this paper, but additional analyses would be helpful. See the Questions section for more details.
2. The presentation could be improved; all figures are quite blurry and lack high quality.
3. The writing, especially in the experiments section, needs clarification. The baseline setup is hard to follow as the authors haven't provided an overview of all compared baselines (e.g., MELO). For instance, it’s unclear why Llama-2-7b wasn’t tested on MELO and no T5 for MEMIT.

**Questions:**

1. An ablation study on each proposed component would strengthen the analysis.
2. Could you provide a visualization of the representation space showing edits, paraphrases, and neighbors before and after editing? This would nicely complement the current analysis.
3. Fig6: LAMA->LLaMA
4. Adding more editing methods in the scaling experiment can help to confirm the robustness of PENME.

---

> ### Author Response · Authors · 2024-11-20
> **Response**
>
> We would like to thank Reviewer UAfd for taking the time to review our work and for the useful feedback. We hope to address the stated concerns below.
>
> **1. The writing, especially in the experiments section, needs clarification. The baseline setup is hard to follow as the authors haven't provided an overview of all compared baselines (e.g., MELO). For instance, it’s unclear why Llama-2-7b wasn’t tested on MELO and no T5 for MEMIT.**
>
> We apologize for the lack of clarity.  As detailed in Appendix B.2, we utilize the EasyEdit library [1] for our evaluations. This library extends the original codebases for model editing approaches, enabling the use of additional models that were not supported previously. However, it is important to note that not all models are currently supported for every approach (e.g., Llama-2-7b for MELO). Additionally, MEMIT is specifically designed to work with decoder-only models, which is why we did not conduct evaluations using the T5 model. Working details for MELO are provided in the Related Work Section of the main text.
>
> **2. Could you provide a visualization of the representation space showing edits, paraphrases, and neighbours before and after editing? This would nicely complement the current analysis.**
>
> We have included the requested visualization in Appendix Section F (*VISUALIZATIONS*). The visualization shows that in model representations edits are closer to their neighbours than to their respective paraphrases. Additionally, edits tend to be closer to other edits, which can result in misfires in similarity retrieval systems. In the representation space of the projection network, we can see that the neighbours are far from the edits and the edits are farther away from each other. This visualization is referenced in Section  6.2 (*DISENTANGLED PROJECTION SPACE*) of the main paper.
>
> **3. Fig6: LAMA$->$LLaMA**
>
> Thank you for pointing that out. We have updated the image to rectify this typo.
>
> **4. Adding more editing methods in the scaling experiment can help to confirm the robustness of PENME**
>
> Conducting model editing methods evaluation across different scaling experiments is computationally intensive. To address concerns related to scalability, we focused our primary experiments on comparing PENME's performance with other editing approaches for 2,000 edits, as shown in Table 1. Notably, experiments in relevant literature typically evaluate performance with up to 1,000 edits [2], [3], [4].
>
> **5. The presentation could be improved; all figures are quite blurry and lack high quality.**
>
> We apologize for the image quality issues and have updated all images to enhance image quality.
>
> **6. An ablation study on each proposed component would strengthen the analysis.**
>
> There are three major hyperparameters in PENME. Edit-to-edit pairings, the margin $m$ in the contrastive loss and the data-driven threshold $\tau$. Abalations regarding edit-to-edit pairings and $\tau$ can be found in Section \S7.1 GENERALIZATION AND LOCALITY of the main text. For $m$ we conduct an ablation study and examine its implications for generalization and locality, utilizing 500 samples from the Counterfact dataset with the GPT2-XL model. The table below presents the results with adjustments to $\tau$ for achieving balanced outcomes for the metrics. Margins between 40 and 80 provide a balanced trade-off between generalization and locality. Notably, locality improves with increasing, which can be advantageous in scenarios where minimizing false matches is critical. We have added the results in Appendix Section D.1.2.
>
>
>
> | **Margin $m$**  | **Threshold Adjustment ($\tau$)**   | **Generalization** | **Locality** |
> |-----------------|----------------------------------|---------------------|--------------|
> | 10              | 0                                | 0.634               | 0.831        |
> | 20              | 3                                | 0.891               | 0.880        |
> | 30              | 6                                | 0.958               | 0.948        |
> | 40              | 8                                | 0.967               | 0.977        |
> | 50              | 11                               | **0.978**           | 0.965        |
> | 60              | 13                               | 0.976               | 0.986        |
> | 70              | 17                               | 0.973               | 0.976        |
> | 80              | 17                               | 0.973               | 0.976        |
> | 90              | 20                               | 0.928               | **0.986**    |

---

> > ### Author Response · Authors · 2024-11-20
> > **References**
> >
> > [1] Peng Wang, Ningyu Zhang, Xin Xie, Yunzhi Yao, Bozhong Tian, Mengru Wang, Zekun Xi, Siyuan
> > Cheng, Kangwei Liu, Guozhou Zheng, et al. Easyedit: An easy-to-use knowledge editing framework for large language models.
> >
> > [2] Tom Hartvigsen, Swami Sankaranarayanan, Hamid Palangi, Yoon Kim, and Marzyeh Ghassemi. Aging with grace: Lifelong model editing with discrete key-value adaptors. Advances in Neural Information Processing Systems
> >
> > [3] Lang Yu, Qin Chen, Jie Zhou, and Liang He. Melo: Enhancing model editing with neuron-indexed dynamic lora. In Proceedings of the AAAI Conference on Artificial Intelligence.
> >
> > [4]  Xiusheng Huang, Jiaxiang Liu, Yequan Wang, and Kang Liu. Reasons and solutions for the decline in model performance after editing. Proceedings of NeurIPS.

---

> > ### Comment · Reviewer_UAfd · 2024-11-24
> >
> > Thanks for your time and effort during the rebuttal. I think you have addressed most of my concern in this paper, especially for the presentation. So I decide to revise my scores accordingly.

---

### Official Review · Reviewer_y63D · 2024-10-31

**Soundness:** 2
**Presentation:** 2
**Contribution:** 3
**Rating:** 5
**Confidence:** 4

**Summary:**

The paper points out that knowledge editing methods based on scoping mechanisms are limited by lexical bias. It proposes using a projector network to decrease the distance between edits and paraphrases, and increase the distance between edits and neighbors to address this issue.

**Strengths:**

* The paper highlights the challenge of lexical bias in knowledge editing, providing valuable guidance for future research in this field.
* The paper proposes using a projector network to enhance retrieval within the codebook, effectively improving the generalization of edits and preventing misfires.

**Weaknesses:**

* The authors didn't fully explain their method. In discussing the construction of key-value memory, they described how to create the keys and set the threshold but didn't explain how to obtain the corresponding values.
* The paper claims that PENME enables faster edit retrieval and simplifies edit removal or updates. However, it lacks supporting experimental evidence.
* The hyperparameter m in the loss function is crucial for the projection network's performance, yet the paper lacks ablation studies on this.
* Many methods were selected for performance comparison, but the authors did not explain why these specific methods were chosen.
* The images in the paper are disorganized and difficult to interpret. Combining multiple experiment results into single images reduces readability.
* The organization of the main text and appendix is unclear. For example, ablation experiments present results for different similarity thresholds for edit-to-edit pairings, but this hyperparameter isn't introduced in the main text, making it hard to understand.
* The paper doesn’t provide detailed explanations of the projector networks, such as their parameter dimensions.

**Questions:**

* Why choose L2 distance in the loss function instead of cosine similarity?
* Why choose cosine similarity for edit-to-edit pairings instead of L2 distance?
* What impact does the hyperparameter m have on the projection network's performance?
* What is the architecture of the projection network? Is it similar to a feed-forward layer in a transformer?
* How is the memory value in the key-value memory obtained?
* The paper proposes two data-driven thresholding schemes. Was Option 1 chosen over Option 2 based on experimental results?
* Why were these methods chosen as baselines in the paper? Is it because they achieved state-of-the-art results on certain metrics or share similarities with PENME?

---

> ### Author Response · Authors · 2024-11-20
> **Part 1**
>
> We would like to thank Reviewer y63D for taking the time to review our paper and we appreciate the detailed comments which we hope to address below.
>
> **1. The authors didn't fully explain their method. In discussing the construction of key-value memory, they described how to create the keys and set the threshold but didn't explain how to obtain the corresponding values.**
> **How is the memory value in the key-value memory obtained?**
>
> The ZsRE and Counterfact datasets provide edit prompts $x_i$ alongside their corresponding new outputs $y_i$, as well as paraphrases $p_{ij}$ and neighbours $n_{ij}$. The keys are projector network representations and corresponding values containing a learned similarity threshold ($\delta$) and the new associated output $y_i$. The threshold for each edit $x_i$ is determined by computing the Euclidean distance between the projector network representations for the edit and its training paraphrases $p_{ij}$ and choosing the maximally far paraphrase distance + $\tau$ where $\tau$ is a hyperparameter. As highlighted in Section 4 (*Projector Editor Networks for Model Editing*), alternative playback mechanisms can be seamlessly integrated with this approach, offering a viable alternative to directly storing the new output information. We describe Key-value Memory in detail in Section 4.2.
>
> **2. The paper claims that PENME enables faster edit retrieval and simplifies edit removal or updates. However, it lacks supporting experimental evidence.**
>
> We conducted an ablation study to evaluate the number of codebook entries and retrieval time, scaling from 50 to 300 edits in increments of 50 samples per experiment. The results, shown below, reveal that both MELO [2] and GRACE [1] demand significantly more codebook entries, leading to slower inference times as the number of edits increases. We have added the results in the Appendix Section  (*COMPARISON SCOPING MECHANISM: PENME VERSUS MELO AND GRACE*).
>
> | **Number of Edits** | **PENME**         |                **MELO/GRACE**          ||
> |---------------------|-------------------|---------------------|---------------------|
> |                     | **Runtime (ms)**  | **Runtime (ms)**    | **Codebook Entries**|
> | 50                  | 0.024 ± 0.003     | 0.316 ± 0.090       | 269                 |
> | 100                 | 0.115 ± 0.129     | 0.364 ± 0.050       | 523                 |
> | 150                 | 0.188 ± 0.182     | 0.624 ± 0.082       | 785                 |
> | 200                 | 0.279 ± 0.170     | 1.423 ± 0.180       | 1048                |
> | 250                 | 0.404 ± 0.170     | 1.681 ± 0.205       | 1319                |
> | 300                 | 0.418 ± 0.125     | 2.149 ± 1.069       | 1554                |
>
> **Caption:** Runtime Performance Comparison of PENME versus MELO retrieval system. For PENME, the number of Codebook entries is the same as the number of edits.
>
>
> **3. The hyperparameter m in the loss function is crucial for the projection network's performance, yet the paper lacks ablation studies on this.**
>
> **What impact does the hyperparameter m have on the projection network's performance?**
>
> We conducted an ablation study on the hyperparameter $m$ for the GPT2-XL model. The table below presents the margin $m$ alongside the corresponding adjustments to $\tau$ to achieve a balance between generalization and locality. Margins between 40 and 80 provide a balanced trade-off between generalization and locality. Notably, locality improves with increasing $m$, which can be advantageous in scenarios where minimizing false matches is critical. We have added the results in Appendix Section D.1.2.
>
> | **Margin $m$**  | **Threshold Adjustment ($\tau$)**   | **Generalization** | **Locality** |
> |-----------------|----------------------------------|---------------------|--------------|
> | 10              | 0                                | 0.634               | 0.831        |
> | 20              | 3                                | 0.891               | 0.880        |
> | 30              | 6                                | 0.958               | 0.948        |
> | 40              | 8                                | 0.967               | 0.977        |
> | 50              | 11                               | **0.978**           | 0.965        |
> | 60              | 13                               | 0.976               | 0.986        |
> | 70              | 17                               | 0.973               | 0.976        |
> | 80              | 17                               | 0.973               | 0.976        |
> | 90              | 20                               | 0.928               | **0.986**    |

---

> > ### Author Response · Authors · 2024-11-20
> > **Part 2**
> >
> > **4. Many methods were selected for performance comparison, but the authors did not explain why these specific methods were chosen.**
> >
> > GRACE and MELO were selected for comparison because they are weight-preserving approaches, similar to PENME, which ensures that model weights are minimally disrupted while integrating new edits. This makes them relevant benchmarks for evaluating the efficiency and precision of PENME. In addition, MEMIT and SERAC were included as they represent high-performing techniques in model editing. MEMIT is a weight-modification-based approach that directly encodes edits by adjusting weights, while SERAC employs external components to handle edits without altering the core model. These diverse approaches provide a comprehensive basis for assessing PENME's performance across different model editing paradigms, highlighting its superior performance. Section  5 (*EXPERIMENTAL SETUP*) has been updated to clarify this. Moreover, the Introduction and Related Work sections provide further details on various types of model editing methods present in the literature.
> >
> > **5. The images in the paper are disorganized and difficult to interpret. Combining multiple experiment results into single images reduces readability.**
> >
> > We apologise for the issues with the figures. We have enhanced the image quality of all figures. Based on the comment, we understand that Figure 6 may be challenging to interpret, as it is the only figure that includes combined ablations on both edit-to-edit pairings $\phi$ and the distance threshold $\tau$. The ablation is presented in a single visualization as it naturally addresses how the two parameters change in combination.
> >
> > To improve the readability of the figure we have increased the iteration gap between $\phi$ and have restricted the visualization to two models (GPT2-XL and T5-small) with the visualization for the last model (Llama-2-7b) being provided in the Appendix Section F (*VISUALIZATIONS*). The paper text and figure caption have been updated accordingly.
> >
> > **6. The organization of the main text and appendix is unclear. For example, ablation experiments present results for different similarity thresholds for edit-to-edit pairings, but this hyperparameter isn't introduced in the main text, making it hard to understand.**
> >
> > We apologise for the omission. We have introduced $\phi$ as the hyperparameter utilized for this pairing. The paper text has been updated to reflect this in Section 4.1 (*PROJECTION NETWORK*) where edit-to-edit pairings are initially introduced.
> >
> > **7. The paper proposes two data-driven thresholding schemes. Was Option 1 chosen over Option 2 based on experimental results?**
> >
> > We decide it based on intuition. However, we added empirical results in the rebuttal to support our intuition. In the following, we summarize our intuition and empirical results.  **Option 2**for an edit may result in a threshold that is lower than the most distant training paraphrase, meaning we can not guarantee generalization for the training paraphrases. For instance, when performing 500 edits on the Counterfact dataset, for a total of 2500 neighbouring pairs $8.62\%$ of the edits encounter this issue.  In contrast, **Option 1** does not prioritize locality for the training neighbours. We have updated Section 4.2 (*KEY-VALUE MEMORY*) to provide clarification.
> >
> > **9. Why were these methods chosen as baselines in the paper? Is it because they achieved state-of-the-art results on certain metrics or share similarities with PENME?**
> >
> > We select a diverse set of model editing methods including both weight-preserving and weight-modified methods. Our motivation to include various diverse methods is to evaluate the effectiveness of our proposed method across all types of methods. However, in terms of methodological similarity, GRACE and MELO are closest to PENME.
> >
> > **10. Why choose cosine similarity for edit-to-edit pairings instead of L2 distance?**
> >
> > The bounded range of cosine similarity simplifies the process of determining a threshold across edits. While L2 distance can also be a valid approach and yields similar results, the goal is to pair edits that are closer in the vector space.
> >
> > **11. What is the architecture of the projection network? Is it similar to a feed-forward layer in a transformer?**
> >
> > The projection network is similar to the feed-forward layers in a Transformer as it contains two layers with ReLU activation in between with an addition of a Batch Normalization layer, a common element in contrastive learning. We have updated Section D (*EXPERIMENTATION AND IMPLEMENTATION DETAILS*) in the Appendix to clarify this.

---

> ### Author Response · Authors · 2024-11-20
> **Part 3**
>
> **12. Why choose L2 distance in the loss function instead of cosine similarity?**
>
> Cosine loss only takes into account the direction of the representations, whereas L2 loss considers both the direction and the magnitude. Existing literature indicates that training with a contrastive objective using L2 loss leads to more separable clusters compared to cosine loss. One issue with cosine loss is that reducing the angle between positive representations increases the magnitude of the representations. Thus, enforcing cosine similarity may result in a leaked heuristic that allows the model to manipulate the magnitude in the projection space, leading to unexpected incompatibility with the downstream computation flow. Furthermore, cosine loss is prone to gradient diminishing effects due to this increase in magnitude, as well as when the initial angle between positive pairs is large. We refer the authors to [3] which outlines these issues. We also provide experimental evidence for these issues by training GPT2-XL on 500 samples from Counterfact, where we modify the objective function of the projector network to cosine loss within PENME’s pipeline. The results show that a negative $\tau$ needs to be set, which means that the paraphrases used during training will fail. Moreover, the performance is lower compared to training with the contrastive learning objective.
>
> ### PENME Cosine Loss
> | **GPT2-XL** |                    |              | **Llama2-7b** |                    |              |
> |-------------|--------------------|--------------|-------------|--------------------|--------------|
> | **$\tau$**  | **Generalization** | **Locality** | **$\tau$**  | **Generalization** | **Locality** |
> | -2          | 0.463              | **0.779**    | -10         | 0.482              | **0.536**    |
> | -1          | 0.691              | 0.603        | -9          | 0.546              | 0.506        |
> | 0           | 0.878              | 0.423        | -8          | 0.575              | 0.466        |
> | 1           | 0.981              | 0.250        | -7          | 0.618              | 0.429        |
> | 2           | **1.00**           | 0.096        | -6          | **0.654**          | 0.392        |
>
> ---
>
> ### PENME Contrastive Loss
> | **GPT2-XL** |                    |              | **Llama2-7b** |                    |              |
> |-------------|--------------------|--------------|-------------|--------------------|--------------|
> | **$\tau$**  | **Generalization** | **Locality** | **$\tau$**  | **Generalization** | **Locality** |
> | 10          | 0.956              | **0.991**    | 10          | 0.935              | **0.99**     |
> | 13          | 0.972              | 0.990        | 13          | 0.963              | 0.987        |
> | 18          | 0.978              | 0.984        | 15          | 0.971              | 0.985        |
> | 19          | 0.980              | 0.980        | 18          | 0.981              | 0.98         |
> | 20          | **0.982**          | 0.975        | 20          | **0.987**          | 0.973        |
>
> [1] Tom Hartvigsen, Swami Sankaranarayanan, Hamid Palangi, Yoon Kim, and Marzyeh Ghassemi.
> Aging with grace: Lifelong model editing with discrete key-value adaptors. Advances in Neural Information Processing Systems
>
> [2] Lang Yu, Qin Chen, Jie Zhou, and Liang He. Melo: Enhancing model editing with neuron-indexed
> dynamic lora. In Proceedings of the AAAI Conference on Artificial Intelligence.
>
> [3] Andrew Draganov, Sharvaree Vadgama, and Erik J Bekkers. The hidden pitfalls of the cosine similarity loss. arXiv preprint arXiv:2406.16468, 2024.

---

> > ### Comment · Reviewer_y63D · 2024-11-24
> >
> > Thank you for your response. However, I believe the following issues remain unresolved:
> >
> > - **Q5 has not been answered**: Is the memory value obtained through training or other methods?
> > - I agree that placing multiple parameters in a single figure can better demonstrate their combined effects. However, without a clear explanation, it might confuse the readers even more. Additionally, the overlapping lines in Figure 5 further increase the difficulty of interpretation.
> > - When changing the margin m, how was the corresponding threshold determined? Is there a quick way to find a suitable threshold?
> >
> > Although I think this paper still has some shortcomings, I also believe that this work effectively improves the generalization of cluster-based knowledge editing methods, which makes it valuable. I am willing to revise my score and hope the authors will continue to refine this paper.

---

> ### Author Response · Authors · 2024-11-24
> **Official Comment by Authors**
>
> **1. Q5  has not been answered: Is the memory value obtained through training or other methods?**
>
> **Response:** Please recall from our previous rebuttal (Part 1, 1) response that the memory value is a tuple consisting of (data driven threshold, $y_i$ the edited output from the dataset), we highlight that any playback mechanism such as vector playback or LoRA indexed blocks can be used as well. The threshold is not learned but is a hyperparameter that is empirically determined.
>
> Thank you for raising this point again, we will ensure this is as clear as possible in the paper as a result of this discussion.
>
> **2. I agree that placing multiple parameters in a single figure can better demonstrate their combined effects. However, without a clear explanation, it might confuse the readers even more. Additionally, the overlapping lines in Figure 5 further increase the difficulty of interpretation.**
>
> **Response** We have updated the caption which now reads *"Figure 5: Shows the trade-off between generalization and locality performance across different hyperparameter settings. The distance threshold $\tau$ varies from $0.01$ to $0.2$ ($0.01$ increments and $\tau$ is normalized by 100), while the edit-pairing similarity threshold $\phi$ ranges from $0.5$ to $0.9$ ($0.1$ increments). Higher $\phi$ values enforce stricter edit similarity requirements. The results showcase the effect of hyperparameter tuning on the projector network's learning capacity and overall performance."*. To make the lines more clearly visible we moved the the minimum value on the y-axis from 0.35 to 0.6.
>
>
> **3. When changing the margin m, how was the corresponding threshold determined? Is there a quick way to find a suitable threshold?**
>
> **Response:** The higher the margin m is kept the higher the value for $\tau$ needs to be. As stated before threshold is not learned but but is itself a hyperparameter that is empirically determined based on the data which in the case of Llama-2-7b is 10 for original experiments (Table 1). A quick way to find a suitable value for $\tau$ is to utilize unseen samples. Using 100 unseen samples we find that 6 is the optimal $\tau$ value for a balanced outcome in generalization and locality for those samples. Utilizing this value for original experiments yields the results shown in the table below. We observe that the scores are high for both generalization and locality but with a stronger emphasis on locality.
>
> | **PENME (Llama-2-7b unseen 100)** ||||
> |-|------------------------|------------------------|------------------------|
> | **τ**  | **Generalization** | **Locality** |
> | 3      | 0.764              | 0.975        |
> | 4      | 0.857              | 0.936        |
> | 5      | 0.875              | 0.904        |
> | 6      | 0.869              | 0.870        |
> | 7      | 0.962              | 0.821        |
> | 8      | 1.0                | 0.828        |
>
> |  **PENME (Llama-2-7b)**  ||||
> |-------------------------|---|---|---|
> | **ES**  | **LOC**  | **PARA** | **Score** |
> | 1.00    | 0.946    | 0.851    | 0.932     |

---

> ### Author Response · Authors · 2024-11-25
> **Official Comment by Authors**
>
> Dear Reviewer y63D,
>
> As the discussion phase draws to a close, we kindly ask whether our response has resolved your concerns or if there are any remaining issues that we can further clarify. Your insights are invaluable in refining our work, and we are eager to ensure that all your concerns are fully addressed.
>
> Thank you once again for your time and effort in reviewing our manuscript.

---

### Author Response · Authors · 2024-11-20
**Central Response**

We would like to thank the reviewers for their valuable comments and feedback. In the following, we reiterate the contribution of our work and summarized the experiments conducted for the rebuttal.

**Novelty:** This is the first work that identifies the issue of lexical bias in model editing. The explicit modelling of a representation space where paraphrases of edits demonstrate proximity while neighbouring examples are far away is a novel solution to the problem at hand. To the best of our knowledge, PENME is the only model editing method designed to maintain both high locality and generalization. The results in Table 1 demonstrate the efficacy of PENME where none of the other methods, whether weight-preserving or weight-modifying, improves both locality and generalization.

**Comparison between PENME, GRACE and MELO:** Reviewer uwxU, y63D and hiiz asked about the difference between PENME and GRACE/MELO. In the following, we compare these methods and have clarified our contribution.

Cluster-based similarity systems like GRACE and MELO [1] and [2] rely on concept separability within the representation space to manually maintain keys in their codebooks. However, our analysis reveals that lexically similar prompts cluster closer to edits than their paraphrases, heightening the risk of system failure as can be seen from figure 1 and 3. Moreover, their cluster based design necessitates storing edit paraphrases as codebook entries for effective generalization which increases retrieval latency. PENME overcomes these limitations by learning a projection space that enhances representation structure, enabling more effective organization of keys for faster and accurate retrieval. Moreover, PENME consistently outperforms both weight-preserving and weight-modifying methods across various architectures, underscoring its adaptability and efficacy.

**Experiments:** We provide a detailed discussion on each experiment in our response to individual comments. In the following, we provide a summary of conducted experiments and their findings.

   1. A runtime comparison between PENME, GRACE, and MELO reveals that PENME is faster due to its requirement for fewer codebook entries. The experiment also highlights that multiple codebook entries are lost during training as a result of cluster resizing operations.

2. Ablation on margin $m$ and the corresponding adjustments needed to $\tau$ for data-driven threshold. The results demonstrate that a value of $40 \leq m \leq 80$ provides a balanced trade-off between generalization and locality.

3. Comparison of cosine loss and contrastive loss for training the projection network. The results demonstrate that the L2 distance performs better than the cosine similarity.

4. Token overlap metrics which showcase the lexical similarity characteristics between edits and neighbours in Counterfact and ZsRE datasets. The results indicate that this issue is not only more pronounced in Counterfact compared to ZsRE but also occurs abundantly within the Counterfact dataset. We also provide data samples from both datasets.

5. For data-driven threshold we provide the percentage of samples where training paraphrases or neighbours fail due to adjustment factor $\tau$. This provides insights for data driven thresholds  Option 1 and 2.

---
**Summary of Revisions:** Details regarding Section numbers are provided in individual comments.
1. The structure of the Appendix has been changed to reflect the order in which it the referred to in the texts.

2. The requested ablations and visualizations have been added to the appendix.

3. Paper components for which further clarifications were requested have been added as blue text in the paper.

4. Image quality of all images has been improved. Figures 3 and 6 have been altered to improve readability.

[1] Tom Hartvigsen, Swami Sankaranarayanan, Hamid Palangi, Yoon Kim, and Marzyeh Ghassemi. Aging with grace: Lifelong model editing with discrete key-value adaptors. Advances in Neural Information Processing Systems

[2] Lang Yu, Qin Chen, Jie Zhou, and Liang He. Melo: Enhancing model editing with neuron-indexed dynamic lora. In Proceedings of the AAAI Conference on Artificial Intelligence.

---

### Author Response · Authors · 2024-11-23
**Comment by Authors**

Dear Reviewers,

Thank you for your thorough feedback on our manuscript. We have addressed all your comments. With the rebuttal deadline approaching, we would greatly appreciate a discussion regarding our responses. Please let us know if there are any points that require further clarification or additional explanation.

Authors

---

### Meta-Review · Area_Chair_ei5w · 2024-12-20

**Metareview:**

This paper explores the tradeoff between generalization (e.g., handling paraphrases) and locality (e.g., avoiding unintended edits on irrelevant queries) in model editing. The authors introduce PENME, a method with two key components: (1) a projection network trained with a contrastive objective to distinguish paraphrased from irrelevant prompts in the representation space, and (2) a memory-based retrieval scheme that improves editing precision by using a similarity threshold as a scoping mechanism. Experiments across three models show that PENME outperforms other baseline approaches.  However, the technical contribution of the work is limited, especially in comparison to previous works like GRACE, where cookbook representations are manually maintained while in this paper, they are learned. Moreover, this paper does not fully explain the methodology, particularly in the construction of key-value memory. The organization and writing of this paper can also be improved. The authors are encouraged to carefully revise the paper based on the reviewers' feedback.

**Additional Comments On Reviewer Discussion:**

The reviewers and authors had discussions, but some reviewers think that this paper lacks  innovation compared to previous baselines such as GRACE.

---

### Decision · Program_Chairs · 2025-01-22

Reject